# Adverse Effects of Increasing Drought on Air Quality via Natural Processes

Yuxuan Wang[1,2]*, Yuanyu Xie[1]*, Wenhao Dong[1], Yi Ming[3], Jun Wang[4], Lu Shen[5]

[1]Department of Earth System Sciences, Tsinghua University, Beijing, China
[2]Department of Earth and Atmospheric Sciences, University of Houston, Houston, TX, USA
[3]NOAA/Geophysical Fluid Dynamics Laboratory, Princeton, NJ, USA
[4]Center for Global and Regional Environmental Research & Dept. of Chemical and Biochemical Engineering & Interdisciplinary Graduate Program in GeoInformatics, University of Iowa, Iowa City, IA, USA
[5]School of Engineering and Applied Science, Harvard University, Cambridge, MA, USA

*These authors contributed equally to this work.

*Correspondence to: ywang246@central.uh.edu; xieyy12@mails.tsinghua.edu.cn*

**Abstract**. Drought is a recurring extreme of the climate system with well-documented impacts on agriculture and water resources. The strong perturbation of drought to the land biosphere and atmospheric water cycle will affect atmospheric composition, the nature and extent of which are not well understood. Here we present observational evidence that US air quality is significantly correlated with drought severity. Severe droughts during the period of 1990-2014 were found associated with growth-season (Mar-Oct) mean enhancements in surface ozone and $PM_{2.5}$ of 3.5 ppbv (8%) and 1.6 μg m$^{-3}$ (17%), respectively. The pollutant enhancements associated with droughts do not appear to be affected by the decreasing trend of US anthropogenic emissions, indicating natural processes as the primary cause. Elevated ozone and $PM_{2.5}$ are attributed to the combined effects of drought on deposition, natural emissions (wildfires, biogenic VOCs and dust), and chemistry. Most climate-chemistry models are not able to reproduce the observed correlations of ozone and $PM_{2.5}$ to drought severity. The model deficiencies are partly attributed to the lack of drought-induced changes in land-atmosphere exchanges of reactive gases and particles and misrepresentation of cloud changes under drought conditions. By applying the observed relationships between drought and air pollutants to climate model projected drought occurrences, we estimate an increase of 1-6% for ground-level $O_3$ and 1-16% for $PM_{2.5}$ in the US by 2100 compared to the 2000s due to increasing drought alone. Drought thus poses an important aspect of climate change penalty on air quality, and a better prediction of such effects would require improvements in model processes.

## 1. Introduction

Air pollution is a major global health risk (Forouzanfar et al., 2015). Chronic and acute exposure to enhanced ozone ($O_3$) and fine particulate matter with diameters less than 2.5 μm ($PM_{2.5}$) has been associated with many adverse health impacts and premature mortality (Lelieveld et al., 2015). Ambient $O_3$ and $PM_{2.5}$ concentrations are strongly regulated not only by the atmosphere but also by land-atmosphere interactions through emission and deposition processes. To date, the variation of air quality with climate change has not been fully revealed as most analysis in the past were conducted with respect to atmospheric parameters or events only, such as temperature (Steiner et al., 2010), precipitation (Dawson et al., 2007; Allen et al., 2015), and short-term (in the order of days) meteorological anomalies (e.g., heat/cold waves, air stagnation, and temperature inversion) (Filleul et al., 2006; Qu et al., 2015; Hou and Wu, 2016). The impact of changing hydroclimate on air pollution is largely unexplored and highly uncertain, particularly with respects to droughts, a type of complex extremes on the time scale of weeks to months or longer which affect not only the atmosphere but also its interactions with the land biosphere.

Drought is characterized by a prolonged period of precipitation shortage and soil moisture deficit in combination with high temperatures (Trenberth et al., 2014). Drought impacts on agriculture and water resources have been extensively documented (Rosenzweig et al., 2001; Arnell, 2004). With regard to air pollution, drought can reduce wet scavenging of pollutants, affect their chemical production/loss, and change their atmospheric lifetime. Drought also influences the health and conditions of soil and vegetation cover across the landscape, thus perturbing upward transmission of dusts (Prospero and Lamb, 2003) and reactive gases (e.g., biogenic volatile organic compounds or BVOCs and $NO_x$) (Fuentes et al., 2000; Guenther et al., 2012; Guenther, 2015; Davidson et al., 2008) from the surface into the atmosphere as well as downward dry deposition of gases and aerosols (Huang et al., 2016). Complications such as increasing wildfires and changing human activities (Westerling et al., 2003; Scanlon et al., 2013) further compound the effects of drought on atmospheric composition. Previous studies illustrated one or two aspects of the complex effects of drought on atmospheric composition. For example, Westerling et al. (2003b) and Prospero and Lamb (2003) explored the potential risks of increasing wildfire and dust emissions under drought conditions. Tian et al (2016) analysed the combined effects of increasing drought and increasing ozone levels on crop production in China, although in their study the ozone change was not linked to drought change. Our prior analysis (Wang et al., 2015) showed a 26% enhancement of surface $PM_{2.5}$ concentrations in the southern US during the severe summer drought conditions in 2011, and suggested wildfires and cloud processes as key factors responsible for the change of $PM_{2.5}$ during drought. A comprehensive assessment of air pollution changes during different drought periods has yet to be conducted to verify the findings from case studies and to reveal important processes responsible for the associated changes. In addition, climate change has the potential to increase the frequency and magnitude of droughts in many parts of the world (Dai, 2012; Cook et al., 2015), further underscoring the importance of understanding the full extent of drought impacts.

In this study we first quantify the impact of historical droughts on air quality, an area largely overlooked in prior investigations of drought impacts, and discuss the possible causes of those impacts. The regional focus is the continental US where observational records of atmospheric composition are most abundant. The study period is 1990-2014 and the growing season (March-October) when drought has most deleterious impacts on the land and biosphere. We then assess the performance of current chemistry-climate models in capturing the response of surface air pollutants to drought. Future changes in air quality related to increasing drought are estimated.

## 2. Data and Method

### 2.1 Drought index

There are many types of drought indices (Heim and Richard, 2002). The drought indices most relevant for air quality would be those capable of representing both meteorological (e.g., temperature and precipitation) and land biosphere conditions (e.g., soil moisture, evapotranspiration, vegetation, etc.) associated with drought, because air pollution levels are dependent not only on meteorology but also the land-atmosphere interaction. In addition, air pollutants have various characteristic time scales, thus the relevant drought indices should be able to specify the duration of droughts. Here we chose the standardized precipitation evapotranspiration index (Vicente-Serrano et al., 2010) (SPEI) to examine the relationship between drought and air quality. The SPEI is multi-temporal in representing the duration of drought and its formulation is based on water balance approach which explicitly considers the impact of temperature variations on evaporation. To identify the full extent of drought impacts and differentiate drought from normal variability in the hydrological cycle, we used the 1-month SPEI to select droughts lasting more than one month. The gridded SPEI datasets are obtained from the global SPEI database (http://sac.csic.es/spei/) with a spatial resolution of $0.5^{o}$ x $0.5^{o}$. While negative SPEI typically indicates drought, a more strict criteria of SPEI < -1.3 (the lowest $10^{th}$ percentile of SPEI) was used here to distinguish drought conditions from non-drought conditions (SPEI between -0.5 to 0.5).

To test the robustness of the drought-pollution relationship derived from SPEI, we used the Palmer Drought Severity Index (PDSI) to evaluate this relationship. The PDSI is the most widely used index of meteorological drought in the US and best represents long-term drought (~12 months) (Heim et al., 2002). Among all forms of PDSI, sc_PDSI_pm is the most updated version with self-calibration and improved formulation of calculating potential evapotranspiration (Dai, 2011). The sc_PDSI_pm dataset (assessed from http://www.cgd.ucar.edu/cas/catalog/climind/pdsi.html) is monthly with a spatial resolution of $2.5^{o}$ x $2.5^{o}$. Drought conditions were identified when sc_PDSI_pm < -3.

Figure 1a shows the percent occurrence of drought months (SPEI < -1.3) over the continental US during the study period. The Western US, Great Plains, Southeast US and southern part of the Northeast US clearly stand out as the most drought prone regions, with extreme droughts occurring 10%-25% of time, ranging between 20 and 40 months during the past 25 years (Figure 1b). Recent examples of infamous droughts are the 2011 Texas drought (Nielsen-Gammon, 2012), the 2012 Great Plains drought (Hoerling et al., 2014), and the 2014-2015 California drought (Griffin and Anchukaitis, 2014). The PDSI-derived drought occurrence frequency (sc_PDSI_pm<-3; Figure S1) shows a similar pattern. However, the areas with more than 10% drought occurrence based on sc_PDSI_pm are much smaller than those based on SPEI (Figure S1a). This is partly because the two indices represent drought at different time scales (i.e. one month for SPEI versus 12 months for sc_PDSI_pm).

### 2.2 Air pollution and meteorological data

Surface concentrations of $PM_{2.5}$ and maximum daily 8 hour running average (MDA8) ozone over the same period were derived from daily observations collected over more than 2000 surface sites from the US Environmental Protection Agency Air Quality System (EPA-AQS) (http://aqsdr1.epa.gov/aqsweb/aqstmp/airdata/download_files.html), Clean Air Status and Trends Network (CASNET; https://www.epa.gov/castnet), and the Interagency Monitoring of Protected Visual Environments (IMPROVE) (Malm et al., 1994, http://views.cira.colostate.edu/) networks. Those daily observations were averaged to monthly means for analysis. The site-specific SPEI is the SPEI at the grid containing each site. Speciated $PM_{2.5}$ data was obtained from the Speciation Trend Networks (STN), which is a subset of the EPA AQS with about 180 sites. Sulfate wet depositions were collected from the National Atmospheric Deposition Program (NADP; http://nadp.isws.illinois.edu/). Isoprene

concentrations were obtained from the Photochemical Assessment Monitoring Stations (PAMS) network (https://www3.epa.gov/ttnamti1/pamsdata.html). Surface data at each site was deseasonalized and detrended by removing the 7-year moving averages from the raw data time series for each month to derive the anomalies (c.f. Figure S2 for an example of data processing). The relationship between SPEI and air pollution anomalies was calculated by linear regression, and the $p$ values are obtained from two-tailed $F$-test. Regional analysis focuses on four geographical divisions of the continental US (Figure 1a): the Western US [128°W-106°W, 30°N-50°N], the Great Plains [106°W-96°W, 25°N-50°N], the Southeast US [96°W-75°W, 25°N-38°N] and the Northeast US [96°W-63°W, 38°N-50°N].

Fire emissions were obtained from the Global Fire Emission Database (GFED) at a 0.25° x 0.25° resolution (Giglio et al., 2013; Randerson et al., 2012; Van der Werf et al., 2010; Akagi et al., 2011; http://www.falw.vu/~gwerf/GFED/GFED4/). The spatial impacts of fire smokes can range from a few kilometers to thousands of kilometers depending on the burning area/intensity, injection height, and transport conditions. Fire emissions from 9 grid points (~40 km) around each surface site were sampled to represent the immediate and transported impacts of fires. Temperature and precipitation were obtained from the Climatic Research Unit (CRU, v3.22), which were also used as input data for global SPEI dataset. Monthly mean cloud fractions from satellite observations were obtained from the Clouds and the Earth's Radiant Energy System (CERES) ISCCP-D1like products at a spatial resolution of 1° x 1° for the period of 2000 to 2014 (Minnis et al., 2011, https://ceres-tool.larc.nasa.gov/ord-tool/jsp/ISCCP-D1Selection.jsp). The site-specific meteorological parameters were retrieved from the grid that contains a surface site.

## 2.3 Models

We evaluated the SPEI-pollution relationships simulated by four models from the Atmospheric Chemistry and Climate Model Intercomparison Project (ACCMIP) (Lamarque et al., 2013) that have archived ozone and $PM_{2.5}$ concentrations: GISS-E2-R, GFDL-AM3, NCAR-CAM3.5, and MIROC-CHEM (Downloaded from http://browse.ceda.ac.uk/). The ACCMIP experiments were forced with observed greenhouse gases concentrations from historical runs. The four models used the same anthropogenic and biomass burning emissions of ozone and aerosol precursors (Lamarque et al., 2010). While anthropogenic emissions were yearly specific, biomass burning emissions were present at the decadal mean without inter-annual variations within a specific decade. Natural emissions were not specified, so the models treated natural emissions differently with different responses to drought. For example, isoprene, the most abundant BVOC, is an important precursor of tropospheric ozone and secondary organic aerosols. Only the GISS-E2-R model simulates isoprene emissions as coupled with its meteorology (mostly temperature), thus allowing for isoprene emissions to increase with increasing temperatures. The other three models used prescribed BVOC emissions, thus representing different responses of those emissions to meteorology and climate change. ACCMIP focuses on time-sliced experiments, thus each model covers different time periods. Model ozone and $PM_{2.5}$ were deseasonalized and detrended for each time slice experiment in order to remove the effect of changes in anthropogenic emissions.

The model SPEI was calculated using the R package provided by the SPEI developers (https://cran.r-project.org/web/packages/SPEI/index.html), with simulated precipitation and temperature from each model as inputs. Model temperature was used to estimate reference evapotranspiration using the simplified Thornthwaite (Th) method. The model SPEI was then derived based on logistic-normalized distribution of water deficit, which is the difference between the reference evapotranspiration and model precipitation. Although more accurate estimates of evapotranspiration can be derived using the more complicated Penman-Monteith (PM) method, as used in the historical SPEI database, it requires additional input data not available from the ACCMIP archive. The Th-derived

SPEI is shown to have tight correlations with the PM-derived SPEI (r > 0.9) (Beguería et al., 2014). The relationship of SPEI with air pollution anomalies was derived over all the time periods with available model outputs. To evaluate cloud properties in the model, we used the random overlap approach (Stephens et al., 2004) to calculate the total cloud fraction (CF) (1000 hPa to 10 hPa) and boundary layer CF (1000 hPa to 800 hPa), which can be the relevant cloud property for tropospheric ozone photochemical formation and cloud processing of aerosols, respectively. Further details on the model experiments and data processing are listed in Supplementary Table S1.

## 3. Retrospective Analysis

### 3.1 Association between drought and air pollution

We first derived the general association of surface ozone and $PM_{2.5}$ with the SPEI at the surface sites.. To remove the effects of seasonality and long-term changes in anthropogenic emissions, pollutant concentrations at each surface site were deseasonalized and detrended (as described in Section 2.2), and the resulting anomalies were used for analysis. Ozone and $PM_{2.5}$ anomalies show spatially prevalent negative correlations with the SPEI (Figure 1), with statistically significant correlations ($p < 0.05$) at 75% - 88% of the sites. Since negative SPEI indicates drought, the negative correlations indicate higher pollution levels during drought. The slopes from linear regression span a range of -4.75~-0.42 for the ozone-SPEI relationship and -2.2~-0.15 for that of $PM_{2.5}$-SPEI, with the mean of -2.21 and -0.83, respectively. The mean regression slope could be interpreted to indicate that an increase in drought severity by one standard deviation of the SPEI is associated with an average increase of 2.21 (±0.85) ppbv for ozone and 0.83 (±0.37) µg m$^{-3}$ for $PM_{2.5}$ in the US. Both slopes are higher in the east, suggesting larger sensitivities to drought for both ozone (-2.63) and $PM_{2.5}$ (-1.0) (Figure 1d and 1g). Consistently with the SPEI, the PDSI shows statistically significant negative correlations with ozone and $PM_{2.5}$ anomalies at the majority of the sites (Figure S1).The correlations with the PDSI are nevertheless weaker, because the SPEI is more suitable to present drought at the monthly scale than PDSI.

To further distinguish the drought effects, we aggregated pollutant anomalies from the sites with greater than 10% occurrence of drought onto three dryness levels: drought (SPEI < -1.3), normal (SPEI between -0.5 and 0.5), and wet (SPEI >-1.3). The composite comparison of ozone and $PM_{2.5}$ between those dryness levels is shown in Figure 1e and 1h, respectively. Significantly higher levels of both pollutants are found associated with drought across all the regions. The eastern sites show a larger enhancement of ozone and $PM_{2.5}$ during drought, consistent with the SPEI-pollutant regression slopes being highest in this region. The response of air pollution to drought can be quantified as the difference of pollutants (ozone and $PM_{2.5}$) anomalies during drought relative to their levels during normal conditions. This difference is referred to as enhancement because it is predominantly positive. The average of such enhancements in the US is 3.5 ppbv for ozone and 1.6 µg m$^{-3}$ for $PM_{2.5}$. Despite regional differences in absolute pollution levels, the relative pollution enhancement during drought is similar across regions at about 8% for ozone and 17% for $PM_{2.5}$ (Figure S3). The enhancements reported hereafter are all evaluated relative to normal conditions unless noted otherwise; if relative to wet conditions, the magnitudes of the enhancements are typically about a factor of two higher.

The composite comparison based on the PDSI is displayed in Figure S1. Since drought frequency represented by the PDSI is comparatively lower, we chose sites with more than 5% drought occurrence and 5 years of available surface observations to reduce the spatial sampling bias. The average enhancement associated with drought derived from the PDSI is 2.3 ppb for $O_3$ and 1.2 µg m$^{-3}$ for $PM_{2.5}$ in the US. The relative enhancement is similar across different regions at about 5% for ozone and 14% for $PM_{2.5}$, smaller but consistent with the results based on the SPEI. Such consistency indicates that the

association between air pollution and drought would not depend on one's choice of drought indicator. However, the SPEI is a more suitable index than the PDSI to identify the drought-pollution association at the monthly time scale, and the following analysis is all based on the SPEI.

To test the robustness of the drought-pollution association and explore the temporal characteristics of such association, we performed the composite comparison of air pollutants between drought and normal conditions separately by season (spring, summer, fall) and by drought stage (onset vs. prolonged). Drought onset is defined as the first month when a drought occurs at a given location; if a drought lasts only one month, that month is also labeled as onset. A prolonged drought is one when the proceeding month is also a drought. Figure 2 compares the variations of regional ozone and $PM_{2.5}$ enhancements during drought derived from different temporal sampling approaches. The growing season (Mar-Oct) mean enhancement of ozone is close to 3 ppbv in the west and Great Plains, increasing to 3.9 ppbv in the southeast and northeast. The same spatial gradient is found in the growing season mean enhancement of $PM_{2.5}$, which increases from a mean of 0.9 μg m$^{-3}$ in the west and Great Plains to 2 μg m$^{-3}$ in the southeast and northeast. Seasonally, all the regions see larger ozone enhancements in summer (Jun-Aug) and fall (Sep-Oct), with the spring (Mar-May) enhancement being the smallest. The Southeast and the Great Plains have the largest seasonal difference in the response of ozone to drought. Relative to the growing season mean, the ozone enhancements in those region are about 38% higher in summer/fall and 50% lower in spring. The seasonal differences of $PM_{2.5}$ enhancements are not statistically significant for most regions, nor are they coherent between regions, probably due to the complexity in $PM_{2.5}$ chemical constituents and sources (to be discussed in Section 3.3). Only the northeast shows a significantly larger $PM_{2.5}$ enhancement in summer and significantly smaller enhancement in spring, about 42% higher and 27% lower than the growing-season mean, respectively. The seasonal comparison for a given region is based on the same sets of surface sites that experienced droughts in all the seasons, thus the differences presented above are not caused by sampling differences.

With respect to drought duration (Figure 2), surface observations in all the regions reveal significantly larger enhancements of both ozone and $PM_{2.5}$ during prolonged drought months than the onset months, with the only exception of $PM_{2.5}$ in the northeast which shows a significantly higher enhancement during drought onset. The largest sensitivity to drought duration is found in the southeast, where both ozone and $PM_{2.5}$ enhancements are higher by up to 50% during prolonged droughts than the onset. Again, the differences shown for a given region are not caused by sampling differences, as the comparison is based on the same sets of surface sites.

**3.2 Meteorological factors for the drought-pollution association**

Before proceeding with a discussion of the causes of the drought-pollution association, it is useful to present the difference between drought and some meteorological conditions/extremes likely to co-occur with drought that are also associated with higher pollution levels. For example, high ozone is more likely to occur with high temperature and low RH (Hou et al., 2016; Zhang and Wang, 2016; Zhang et al., 2017), and during heat waves (Filleul et al, 2006), conditions often co-occurring with drought. High $PM_{2.5}$ events in the US are found to co-occur with high temperature and low wind speed, but not consistently dependent on RH (Tai et al., 2010; Zhang et al., 2017). Stagnation days typically result in high ozone and $PM_{2.5}$ levels at the surface (Tai et al., 2010; Schnell and Prather, 2016). Compared with those types of weather phenomena and extremes defined on the daily basis, drought has on a longer time scale of at least one week and often monthly. For example, one would not call it a drought if no rain for a few days. Drought arises only after a prolonged (> week) period of precipitation shortage that causes soil to dry up. Therefore, we chose the monthly scale to identify the drought-pollution association, differentiating it from day-to-day variability of meteorology.

Furthermore, drought is a complex extreme not based on individual meteorological parameters (e.g. temperature, humidity) or a simple combination of them. The prominent feature of drought is water deficit in both the atmosphere and the land component (e.g. soil and vegetation), resulting from the combination of precipitation shortage and increasing evapotranspirative water loss driven in part by high temperatures. During the historical drought periods analyzed here, the surface sites were affected with precipitation decreases of up to 50% regionally and temperature increases of up to 2 $^{o}$C, as compared to normal conditions (Table S2). Large changes in other meteorological variables are also associated with drought conditions, such as a 10% decrease in RH, 39% decrease in cloud fraction, and an increase in incoming solar radiation by 12.4 W/m$^2$. Because the time scale of drought is monthly, these meteorological changes are persistent changes on the monthly scale, as opposed to day-to-day variability. As a result, the associated vegetation responses are likely to be more pronounced during drought than those associated with short-term meteorological extremes/events, with important implications for the land-atmosphere exchanges of reactive gases and aerosols.

As discussed above, there are well-established linkages between air quality and some meteorological parameters (e.g. temperature), thus the drought-pollution association may be partly explained by the effects of drought on these meteorological variables. For example, the co-occurrence of drought with high temperature and low RH is an important reason to explain the pollutant enhancements during drought, especially for surface ozone. However, it would not be feasible to separately quantify the effects of certain meteorological variables on the drought-pollution association, such as temperature, precipitation, and RH, because these variables are all factored in when defining drought. But other meteorological variables might be ruled out as compounding the drought-pollution association. For example, wind speed is a key factor influencing air quality, but not an explicit factor in drought indices. The correlations (r) of monthly mean wind speeds from ERA-reanalysis with the SPEI (Figure S4) are positive but small for the most part of the US ($r^2$ < 0.2), except for the northwest corner and surroundings of the Great Lakes with $r^2$ of 0.3~0.4. This suggests that wind speeds might not be an important meteorological factor responsible for the pollution enhancements during drought, except for localized areas where wind-blowing dust would become substantially higher during drought.

In addition, drought months may consist of a larger number of meteorological extremes conducive for high pollutant levels, such as stagnation and heat waves. To understand the pattern and extent of such co-occurrence, we examined the relationships of monthly occurrences of stagnation and heat waves with the SPEI at each 0.5$^o$x0.5$^o$ grid over the study period (Figure S4). The frequency of stagnation was derived from the NOAA Air Stagnation Index (ASI, https://www.ncdc.noaa.gov/societal-impacts/air-stagnation/), in which stagnation is defined when surface wind speed smaller than 3.2 m/s, 500 mb wind smaller than 13 m/s and no precipitation, following Wang and Angell (1999). Heat waves were defined as two consecutive days with daily mean temperature greater than 90$^{th}$ percentile of the warm-season (May to Sep) daily mean temperature during 1990-2014, following the method by Anderson (2011), using ERA Interim reanalysis (Dee et al., 2011) as inputs. Temporally, both stagnation and heat waves show negative correlations with the SPEI across the US (upper panel of Figure S4), an indication of more days of these meteorological extremes during drought months, but the squares of these correlations are all below 0.4, with a typical value of 0.1-0.2 for the most parts of the US. This suggests that on the monthly scale stagnation and heat waves would typically be able to explain 10%~20% variability in the SPEI, a non-trivial but small fraction. The exceptions are found in isolated locations in the west and southeast where stagnation could explain up to 40% of the SPEI variability, and the southern Great Plains with up to 30% of the SPEI variability explained by heat waves. Stagnation has an overall higher correlation with the SPEI than heat waves, partly because stagnation days by definition exclude precipitation. The lower panel of Figure S4 shows that stagnation and heat waves have an average 7% and 5% increase in their frequencies during drought months compared to normal months, although the extent of such increases varies greatly by region. The

maximal increase of stagnation frequency during drought is about 15% in the west, southern Great Plains and southwest, where stagnation tends to occur frequently even during normal conditions. The largest increase of heat waves during drought is about 20% in the southern Great Plains.

To quantify the compounding effects of stagnation and heat waves on the drought-pollution association, we reevaluated the SPEI-pollutant relationships by applying weights to each pair of SPEI and pollutant anomalies (ozone and $PM_{2.5}$). The weights are given as the percentages of days in each month (regardless of drought or non-drought) that are neither stagnation nor heat wave, assuming the two events are mutually exclusive which would give an upper bound for the weights. For example, a month with none of the two events is given 100% weight when calculating the SPEI-pollutant correlation and pollutant enhancement, while a month with 15 days of those events has a weight of 50%. The weighted enhancement is calculated as the difference in weighted-mean anomalies between drought and normal months. Since the weights are between 0 and 1, the weighting process effectively scales down the magnitude of pollution anomalies in each month, assuming the effects of stagnation and heat waves are linear to their occurrences. Figure 3 compares the original (un-weighted) and weighted correlations, regression slopes, and pollution enhancements. The differences in correlation coefficient (r) are mostly smaller than 0.05 in terms of absolute values. The exception is for ozone in the west where the absolute value of the weighted r is increased by 0.1~0.2, revealing a stronger correlation between SPEI and ozone after accounting for the impacts from stagnation and heat waves. The reason why the direction of the correlation changes after weighting can be either an increase or decrease is because the weights are assigned to both drought and non-drought months. The weighted enhancements of ozone are 30-59% lower than the original, un-weighted values, but remain to be significantly positive. The corresponding reduction for the $PM_{2.5}$ enhancements is 27%-45%. The west and southeast have a larger reduction in the enhancements of both pollutants after weighting, consistent with the fact that these regions show a larger increase in stagnation and heat wave frequencies during drought. The same weighting method can be separately applied to stagnation and heat waves to compare their effects individually (Figure S5-6). Stagnation exerts a larger influence on the weighted enhancement in the west and southeast, while heat waves has a larger effect in the Great Plains, consistent with the spatial distribution of their respective occurrences during drought. In all the cases examined here, the weighting does not change the sign or statistical significance of the SPEI-pollutant correlations at all the sites, indicating the covariance of drought with stagnation and heat waves might not be the dominant factor causing the SPEI-pollutant correlations. The weighting however reduces the magnitude of ozone and $PM_{2.5}$ enhancements associated with drought in every region, with an average reduction of 40% when both events are counted together as weights. This indicates that more frequent stagnation and heat waves could explain up to 40% of the ozone and $PM_{2.5}$ enhancements during drought, a significant but not majority factor.

**3.3 Emission/deposition/chemistry factors for the drought-pollution association**

Drought can further affect air quality through perturbations to emissions, deposition, and chemical processes. High temperature conditions during drought will lead to higher production rate of ozone as well as higher emissions of BVOCs (Fuentes et al., 2000; Guenther et al., 2012). Surface observations of isoprene suggest 7-20% higher concentrations under drought conditions (Table 1; Figure 4). An exception is a decrease in isoprene during severe drought (SPEI < -2) over the southeast and northeast US (Figure 4), presumably due to shutoff of isoprene emissions when severe water stress causes reduction in carbon sources, lower level of isoprene synthase gene expression, stomata closure and wilting of vegetation (Pegoraro et al., 2004; Brilli et al., 2007; Seco et al., 2015). Surface $NO_2$ was found to be higher by 0.07-1.26 ppb (2-9%), attributable to increased emissions from fires, soils and possibly the power sector. Precipitation scavenging of air pollutants should be much lower during drought, resulting in higher pollutant concentrations and longer lifetime in the atmosphere. Compiling the scattered measurements by the National Atmospheric Deposition Program, we found a 23-32%

reduction in wet deposition of sulfate during drought (Table 1). In addition, severe drought can potentially lead to elevated surface ozone by reducing the ozone dry deposition to vegetation (Fowler et al., 2009; Kavassalis and Murphy, 2017). A modelling study suggested up to 20% reductions in ozone dry deposition due to lower stomatal conductance during a drought event in Texas (Huang et al., 2016). However, changes of wet and dry deposition fluxes due to drought are difficult to quantify due to dearth of deposition measurements.

The enhancements of $PM_{2.5}$ species during drought are presented in Figure 5 at a subset of surface sites with speciation measurements. Organic aerosol (OA), sulfate and dust are major contributors to the overall $PM_{2.5}$ enhancements. There is a 2-15% increase in sulfate, attributable in part to reduced wet deposition. While oxidation rate of $SO_2$ increase at high temperatures (Tai et al., 2010), surface $SO_2$ shows a 1-10% increase during drought, presumably due to reduced dry or wet deposition and higher emissions from fires and electricity generation (Scanlon et al., 2013). Dust enhancements are most significant in the west (27%) and the Great Plains (16%) due to more semi-arid areas. Significant OA enhancements (12-35%) are found associated with drought across all the US. Fire emissions of primary OA are 1-3 times higher during drought, explaining a large portion of the OA enhancement (Tables S3-4). When excluding the fire influences, an increase in the OA to BC ratio was found under drought (Figure S8), indicative of an increase in secondary organic aerosols (SOA) formation. However, routine networks provide only limited classification of OA and cannot fully distinguish the response of SOA to drought from that of total OA.

The above analysis suggests that the ozone and $PM_{2.5}$ enhancements during drought are largely responses of natural processes from the land biosphere and abnormal atmospheric conditions. To compare the drought-related changes with the effects of anthropogenic emission reductions in the US, we divided the data into two sub-periods: 1990 to 2003 (P1) and 2004 to 2014 (P2). Anthropogenic emissions of $PM_{2.5}$ and ozone precursors have decreased significantly in the US from P1 to P2, for example, by 50% for $SO_2$ and 32% for $NO_x$ according to the Air Pollutant Emission Trend Data from the US EPA (EPA, 2016). In spite of this, drought-related enhancements of ozone and $PM_{2.5}$ are manifested clearly in both periods, with little change in the magnitude of these enhancements between P1 and P2 (Table 2). Under normal conditions, there is a decrease of ozone and $PM_{2.5}$ from P1 to P2 by an average of1.6 ppbv and 1.8 μg m$^3$, respectively, which is attributable to the reductions of US anthropogenic emissions. By comparison, drought-related mean enhancement of ozone exceeds 4 ppbv in both periods and that of $PM_{2.5}$ is 1.6 μg m$^3$. Therefore, the pollutant enhancements associated with droughts do not appear to be affected by the decreasing trend of US anthropogenic emissions, indicating natural processes as the primary cause.

**3.4 Modeled response of air pollutants to drought**

Previous studies suggest that climate models have some skills to predict the variability of drought (Dai, 2012). Indeed the four models from ACCMIP all reproduce the observed spatial patterns of historical droughts in the US (Figure S7). Simulated severe droughts (model SPEI < -1.3) occur ~20% of the time over the west and southern US, consistent with the observed SPEI. However, the temporal correspondence (i.e. month-to-month) between model SPEI and observed SPEI dataset is weak, largely due to the models' deficiency in simulating temporal variability of precipitation. This weak correlation however is not expected to affect the evaluation of simulated pollution responses to drought, because we used the model SPEI to derive the SPEI-pollutants relationships from each model.

The models vary greatly in their ability of predicting the drought-pollutants relationships. With respects to surface ozone, all the models are able to capture its negative correlation with SPEI over most of the US (Figure 6), as they all predict some levels of increase in ozone production driven by higher temperatures during drought (Figure S9). However, the simulated slopes and magnitude of ozone

enhancement are less than half of the observed values in many regions, suggesting a lack of full representation of the drought effects. The GISS-E2-R model, which is the only model that includes interactive isoprene emissions with model temperature, reproduces the observed isoprene increases. This allows the model to simulate ozone enhancements resulting from stronger isoprene emissions (Schnell et al., 2016), and thus the GISS model simulates the greater SPEI-ozone slope as compared to other models (Figure 6). In spite of lacking the interactive isoprene emissions, the MICRO-CHEM model shows higher ozone enhancements than other models because it simulates the largest increase of ozone production caused by drought, presumably due to a larger sensitivity of ozone to temperature. Drought perturbation of the land biosphere would lead to reductions in the ozone dry deposition sink and hence higher ozone enhancements. For example, a model sensitivity study by Lin et al. (2017) showed that reducing ozone dry deposition velocity by 35% in the GFDL-AM3 model during the severe North American drought of 1988 would lead to 10 ppbv greater ozone enhancements than a simulation with constant dry deposition velocity. However, all the ACCMIP models examined here simulate little changes of ozone dry deposition (-3~5%) during drought.

The models are less skillful in reproducing the effects of drought on $PM_{2.5}$ (Figure 7). All the models incorrectly predict a decrease of $PM_{2.5}$ under drought conditions and hence a positive $PM_{2.5}$-SPEI relationship for many regions in the US, whereas this relationship is clearly negative in the observations across all the regions. For the few regions where some models are correct about the direction of the $PM_{2.5}$ change (e.g. the western US by GISS-E2-R and eastern US by NCAR-CAM3.5), the magnitude of the $PM_{2.5}$ change is less than 70% of that observed.

The model response is primarily driven by a ubiquitous and excessive decrease of sulfate under drought conditions caused by large reductions of sulfate production in clouds (-22~-73%) (Figure S10-11). In contrast, only 14-34% of the sites in the west and the Great Plains show a decrease of sulfate during drought. The model deficiency in sulfate can be explained by their underestimate of low-altitude cloud fraction at higher temperatures (Shen et al., 2016). This bias would lead to an underestimate of sulfate production as well as SOA-processing in clouds during drought (i.e. high temperature conditions), which could outweigh the aerosol deposition decrease. Figure 8 compares the satellite-derived sensitivity of total and boundary-layer CF to drought severity with that simulated by the GISS and GFDL model, which are the only ACCIMP models that archived layer-specific CF. For the boundary layer CF which should be more relevant for in-cloud processing of aerosols, the observed sensitivity averages about 0.51 per unit increase of SPEI, while the GISS and GFDL model shows a sensitivity of 4.37 and 3.41, respectively, about a factor of 8 higher than the observed value. The models also overestimate the sensitivity of total CF to drought, but to a less extent. Another important aspect of model deficiencies is that they all underestimate the OA enhancements in every region. Simulated OA changes primarily result from reduced wet deposition (~40%), lacking important contributions from changing BVOCs emissions, fires, or chemistry (Figure S10-11) as suggested by observations.

In summary, the model deficiencies suggest a lack of mechanistic understanding of natural processes of importance to atmospheric composition and/or their perturbations by drought, although attribution of the underlying causes would require chemistry-climate model sensitivity experiments, which is outside the scope of the present study. Emissions, deposition, and chemistry are the most important aspects of model configurations affecting the drought-pollutants relationship. Since natural emissions were not specified, the ACCIMP models treated natural emissions differently, which is a key factor in the different performance between models. Using the observed SPEI-pollutants relationship as a diagnostic, we found that the model with interactive isoprene emissions (e.g. the GISS model) has a better ability to simulate the SPEI-ozone relationship, indicating the importance of drought effects on BVOCs emissions. With regard to deposition, all the models simulate some levels of decreasing wet deposition during drought, but dry deposition is largely insensitive to drought due to the lack of drought effects on the

properties of the land and biosphere. The overestimate of the dry deposition sink during drought may be another reason behind the models' deficiency in underestimating the drought-pollutants relationship. Lastly, all the models overestimate the sulfate reduction, but at the same time underestimate the OA increase during drought. Both problems might be caused by the model misrepresentation of cloud sensitivity to changing drought severity, although the OA bias could also be caused by uncertainties of fire and BVOC emissions in the model.

## 4. Future changes in drought and adverse impacts on air quality

To circumvent the model deficiencies, the effects of future increases of drought on air quality were estimated by extrapolating their present-day relationships from observations to model projected drought occurrences under future warming scenarios. Projected changes in SPEI from the present to future climate were derived from the outputs of the four ACCMIP models (i.e. GISS-E2-R, GFDL-CM3, CCSM4, and MIROC-ESM-CHEM) archived by the Coupled Model Intercomparison Project Phase 5 (CMIP5) (Taylor et al., 2012). The CMIP5 historical runs cover the period from 1850 to near present, and are forced with observed changes in atmospheric composition with evolving land cover. The future projection runs span from 2006 to 2300, forced with specified concentrations of certain atmospheric constituents defined in three representative concentration pathways (RCPs) scenarios (Moss et al., 2010): RCP 2.6 (low mitigation emission scenario), RCP 4.5 (midrange mitigation emission scenario) and RCP 8.5 (high emission scenario). Changes in future drought conditions compared to the present are defined as the 2100 SPEI (2080-2099 mean) minus its value in 2000 (1990-2005 mean).

Figure 9a shows the projection of SPEI in the US by 2100 (2080-2099 mean) under different RCPs that are derived from the mean of the four models from the CMIP5 outputs. Drought risks are projected to increase with warming scenarios over all parts of the US, with the largest increases in the west and the Great Plains, consistent with previous projections (Cook et al., 2015). These projected SPEI changes (2100 minus 2000), when multiplied by the present-day relationship between SPEI and air quality derived from observations (c.f. Figure 1), suggests a 0.3-3.0 ppb (1-6%) increase of surface ozone and 0.1-1.0 $\mu g\ m^{-3}$ (1-16%) increase of $PM_{2.5}$ in the US in 2100 as a result of increasing drought alone under different RCPs (Figure 7b-c). The increase of ozone and $PM_{2.5}$ are largest in the west. The maximum increase is 14% for ozone and 41% for $PM_{2.5}$ under the extreme warming scenario (RCP 8.5), significantly higher than the present-day effects. While this extrapolation-based projection may not be reliable quantitatively, it suggests a significant climate change penalty on air quality through drought, which has been overlooked before and pose a new challenge for air quality managers.

## 5. Discussion

The retrospective analysis of observations demonstrates that past droughts have been associated with significant deterioration of air quality through natural processes, resulting in potentially large tolls on public health that have not been considered in previous impact analysis of drought. The land biosphere plays a key role in mediating drought-related changes in atmospheric chemistry. The magnitude of the land biosphere response is largely dependent on concurrent changes in solar irradiance, temperature and water at different levels of drought severity and duration. More sunlight and higher temperatures may outweigh some levels of water stress, resulting in enhanced BVOCs emissions through leaf biochemistry, vapor pressure difference and underlying metabolism processes (Fuentes et al., 2000). However, extreme and/or prolonged drought conditions with severe water stress coupled with very high temperatures can affect the activity of enzyme and health of the plants, therefore leading to reductions in BVOCs emissions. More comprehensive understanding of the land biosphere responses is required to quantify the impact of land biosphere to atmospheric compositions under different drought conditions.

In addition to changing BVOCs emissions, reduced aerosol water content under drought conditions can perturb aqueous phase formation of SOA from BVOCs, but the impact is not clear (Gilardoni et al., 2016). Changes in anthropogenic emissions under drought conditions are also uncertain. Local land use type and water management policy can significantly affect human reactions to drought. Furthermore, the interaction between anthropogenic emissions and natural responses further compound the drought effect, as anthropogenic emitted gases and aerosols can affect the oxidation and partitioning processes of SOA from BVOCs (Hoyle et al., 2011; Xu et al., 2015).

Changes in the land biosphere and atmospheric compositions, including gases and aerosols, can provide feedbacks to the climate through radiative effects and cloud interactions. Reductions in vegetation cover affect surface albedo and dust emissions, resulting in enhanced surface temperatures, intensification of drought conditions and geographical shift of drought pattern (Cook et al., 2009). Increasing wildfire activity and fire-emitted aerosols alter the regional energy budget and circulation, which lead to reduced precipitation thus further enhancing drought severity and vulnerability of ecosystem towards wildfires (Bevan et al. 2009; Tosca et al., 2010; Hodnebrog et al., 2016). Improvements in climate-chemistry models are thus imperative to facilitate better prediction of atmospheric composition changes due to changes in drought and improved understanding of the associated feedbacks of composition changes to climate and drought itself.

The observational analysis presented here indicates significant changes of air pollutants under drought conditions. However, it is not sufficient to quantify the full extent of the cascading effects of drought on the complex chemistry of ozone and SOA, which would require more targeted measurements providing for example more classification of organic materials and modeling at the process level. Uncertainties exist in the model assessment since we are using a single version of simulation for each model and the study period is relatively short and may not represent the full simulation results. Nonetheless, both observations and model indicate the important role of the land biosphere and atmospheric conditions in regulating pollutant levels under drought conditions. Future air quality management should consider the adverse effects from increasing drought risks.

## Data availability

All datasets used in this study are publically accessible.

## Author contribution

Y. W. and Y. X. conceived the research idea. Y. X. and W.D. performed the analysis and Y. W. wrote the initial draft of the paper. All authors contributed to the interpretation of the results and the preparation of the manuscript.

## Competing financial interests

The authors declare no competing financial interests.

## Acknowledgements

We thank the individuals and groups involved in making observations at IMPROVE, US EPA, CASNET, NADP and PAMS networks, and in preparing the SPEI, PDSI, GFED, ASI, CRU, ERA-Interim and ISCCP-D1like database. We thank the modelling groups that participate in ACCMIP and CMIP5 for producing the model outputs and making them available. This work was supported in part by the National Key Basic Research Program of China (2013CB956603 and 2014CB441302). J. Wang acknowledges the support from NASA Aura Science program (grant #: NNX14AG01G managed by Dr. Ken Jucks), Applied Science Program (grant #: NNX15AC28A managed by Dr. John Haynes), and ACMAP program (grant #: NNX15AC30G managed by Dr. Richard Eckman).

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

**Tables**

**Table 1: Changes in the concentrations of atmospheric gaseous compositions and sulfate wet deposition under drought compared to normal conditions. Data are from different measurement networks (see Method).**

| | | West | Great Plains | Southeast | Northeast |
|---|---|---|---|---|---|
| $NO_2$ (ppb) | $N^a$ | 130 | 27 | 81 | 122 |
| | Diff | 1.26 (+9.0%) | 0.07 (+2.3%) | 0.14 (+2.6%) | 0.46 (+3.9%) |
| | p-value[b] | <0.01 | 0.68 | 0.25 | <0.01 |
| $SO_2$ (ppb) | N | 66 | 28 | 113 | 290 |
| | Diff | 0.14 (+2.6%) | 0.13 (+1.4%) | 0.29 (+10.4%) | 0.32 (+7.3%) |
| | p-value | 0.05 | 0.28 | <0.01 | <0.01 |
| Isoprene (ppb carbon) | N | 8 | 14 | 28 | 21 |
| | Diff | 0.21 (+11.6%) | 0.01 (+7.0%) | 0.09 (+13.8%) | 0.36 (+19.5%) |
| | p-value | 0.04 | 0.60 | 0.09 | 0.01 |
| Sulfate wet deposition (kg $month^{-1}$) | N | 48 | 30 | 47 | 83 |
| | Diff | -0.62 (-31.7%) | -1.39 (-26.7%) | -2.47 (-22.9%) | -2.99 (-26.2%) |
| | p-value | <0.01 | <0.01 | <0.01 | <0.01 |

a. Number of sites.
b. P value derived from student t-test.

**Table 2: Changes in the concentrations of ozone and PM$_{2.5}$ at two periods under drought compared to normal conditions.**

| | P1 (1990-2003) | | | P2 (2004-2014) | | | P2 minus P1 |
|---|---|---|---|---|---|---|---|
| | Drought | Normal | Diff | Drought | Normal | Diff | Normal |
| **Ozone (ppbv)** | | | | | | | |
| West | 56.61 | 51.58 | 5.03 | 53.19 | 49.15 | 4.04 | 2.43 |
| Great Plains | 51.64 | 47.23 | 4.41 | 52.23 | 47.75 | 4.48 | -0.52 |
| Southeast | 51.98 | 47.01 | 4.97 | 49.03 | 44.75 | 4.28 | 2.26 |
| Northeast | 51.64 | 46.43 | 5.21 | 48.19 | 44.23 | 3.96 | 2.20 |
| Average | 52.97 | 48.06 | 4.91 | 50.66 | 46.47 | 4.19 | 1.59 |
| **PM$_{2.5}$ ($\mu$g m$^3$)** | | | | | | | |
| West | 6.57 | 5.56 | 1.01 | 5.84 | 4.74 | 1.10 | 0.82 |
| Great Plains | 7.69 | 6.22 | 1.47 | 6.86 | 5.81 | 1.05 | 0.41 |
| Southeast | 15.98 | 14.19 | 1.79 | 13.79 | 11.45 | 2.34 | 2.74 |
| Northeast | 16.37 | 14.25 | 2.12 | 12.67 | 10.94 | 1.73 | 3.31 |
| Average | 11.65 | 10.06 | 1.60 | 9.79 | 8.24 | 1.56 | 1.82 |

**Figures**

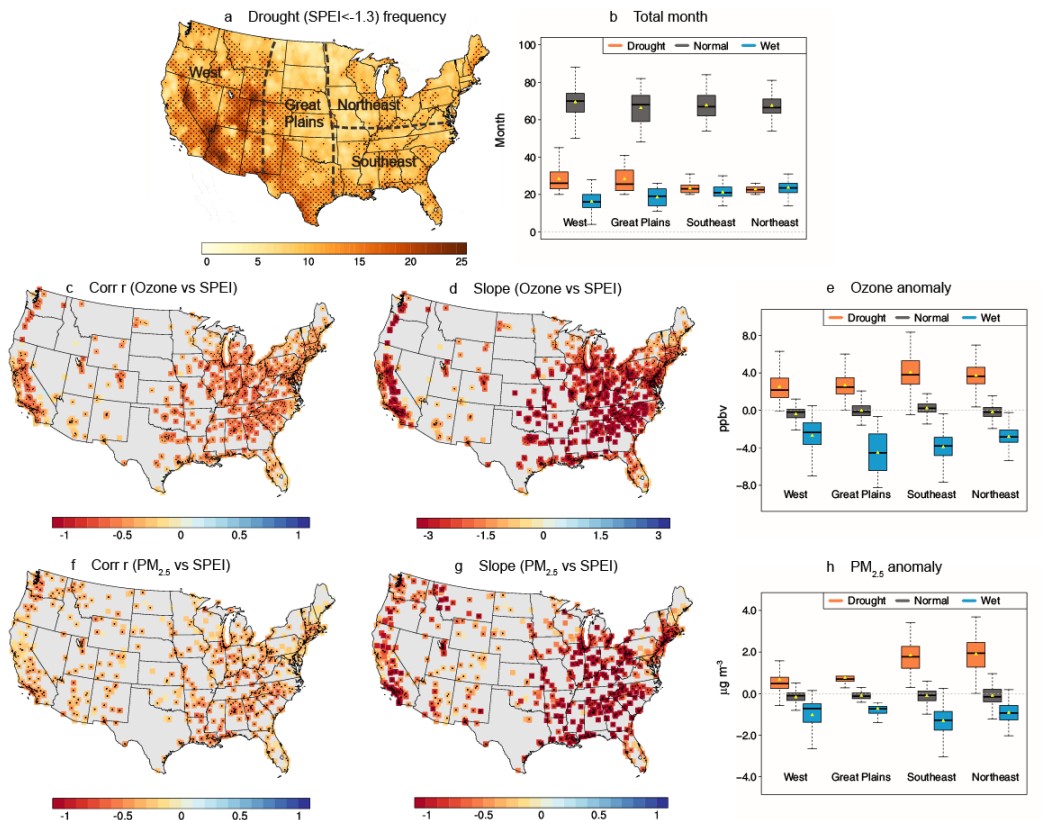

**Figure 1.** (a) Percentage occurrence of severe drought months (SPEI < -1.3) over the continental US during 1990-2014; black dots indicate drought frequency greater than 10% and dashed lines show the four geographical regions. Linear regression correlation coefficient r and slope of SPEI with $O_3$ (c,d) and $PM_{2.5}$ (f,g) anomalies at surface sites with data records longer than 5 years; yellow dots indicate regression significance at 95% confidence level. Boxplot comparisons of total months in different dry sectors (b), ozone (e) and $PM_{2.5}$ anomalies (h) under drought (SPEI < -1.3), normal (-0.5< SPEI < 0.5) and wet conditions (SPEI > 1.3) by region; the yellow triangles in the boxplot indicate mean values. All the surface data shown in the boxplot are restricted to sites with data records longer than 5 years and more than 10% occurrence of severe drought (SPEI < -1.3).

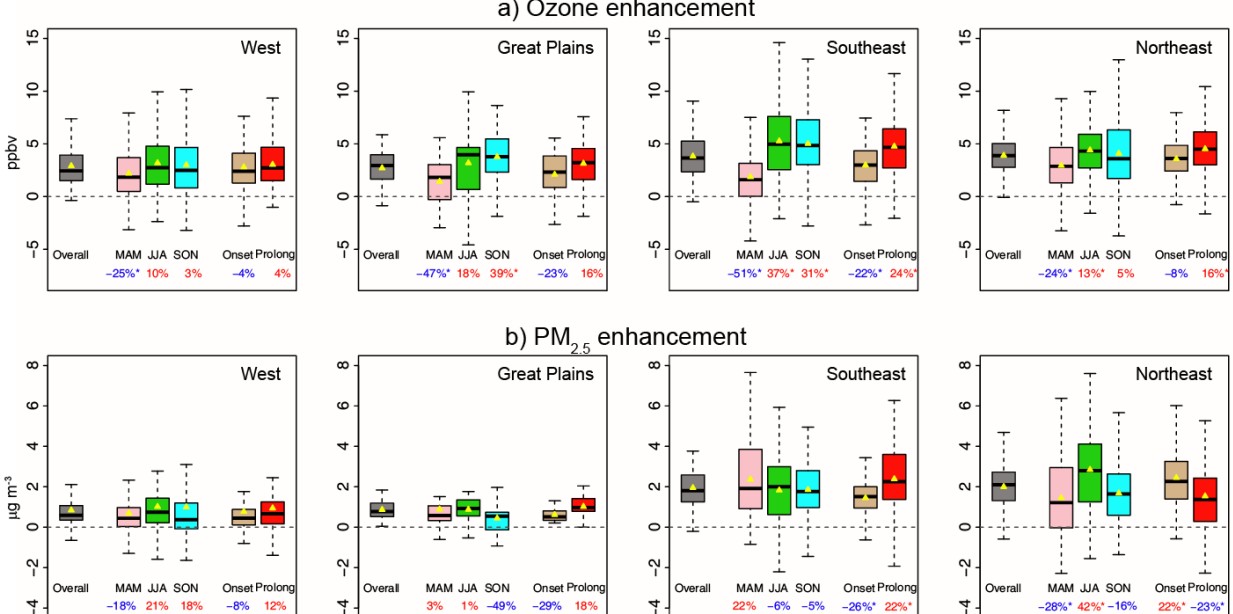

**Figure 2**. Ozone and PM$_{2.5}$ enhancements during drought relative to normal conditions at different seasons and different drought stages. The yellow triangles in the boxplot indicate mean values. The numbers below each box represent the difference relative to the overall enhancement of the whole growing season (Mar-Oct) (grey box), with the asterisks indicating significant differences at 95% level from the Student's t-test

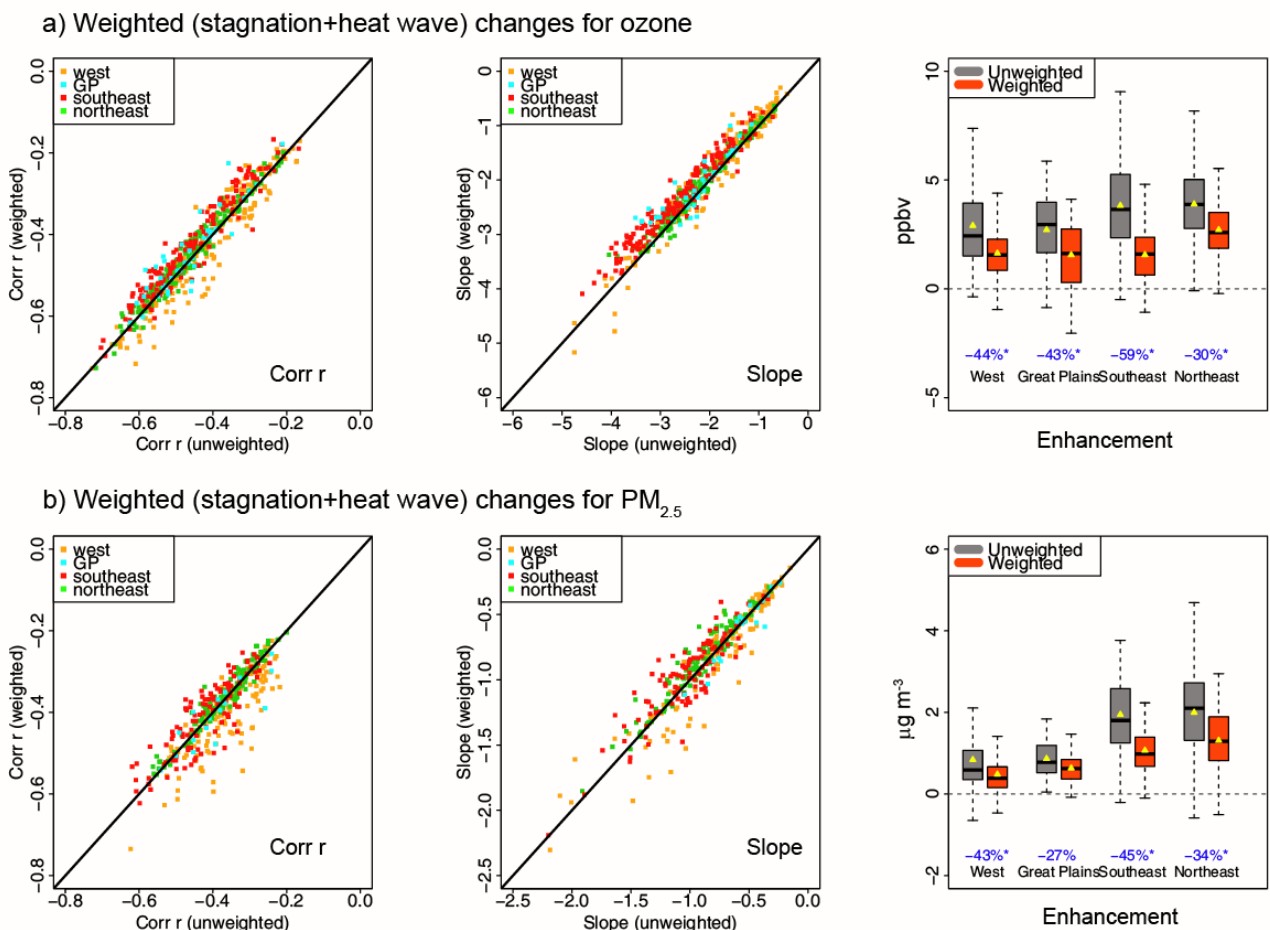

**Figure 3**. Comparison between the weighted (by frequency of stagnation and heat waves) and un-weighted SPEI-pollutants relationship (correlation r, left panel; correlation slope, right panel) and pollutants enhancements (right panel). The upper panel is for ozone and the lower panel for $PM_{2.5}$. Left and middle panels: the black lines are the 1:1 lines and different colors represent different regions. Right panel: the numbers below each box indicate the difference relative to the un-weighted enhancements.

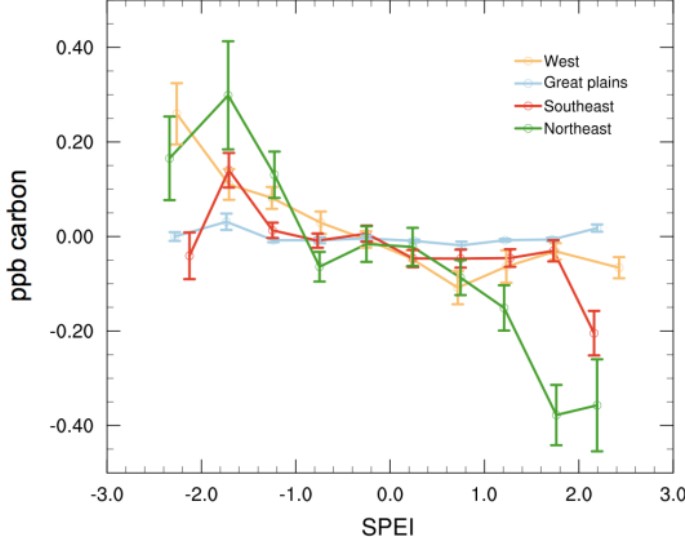

**Figure 4:** Isoprene anomalies (ppb carbon) derived from the PAMS network at binned SPEI levels over the Western, the Great Plains, the Southeastern and Northeastern US. Error bars indicate standard error of the mean.

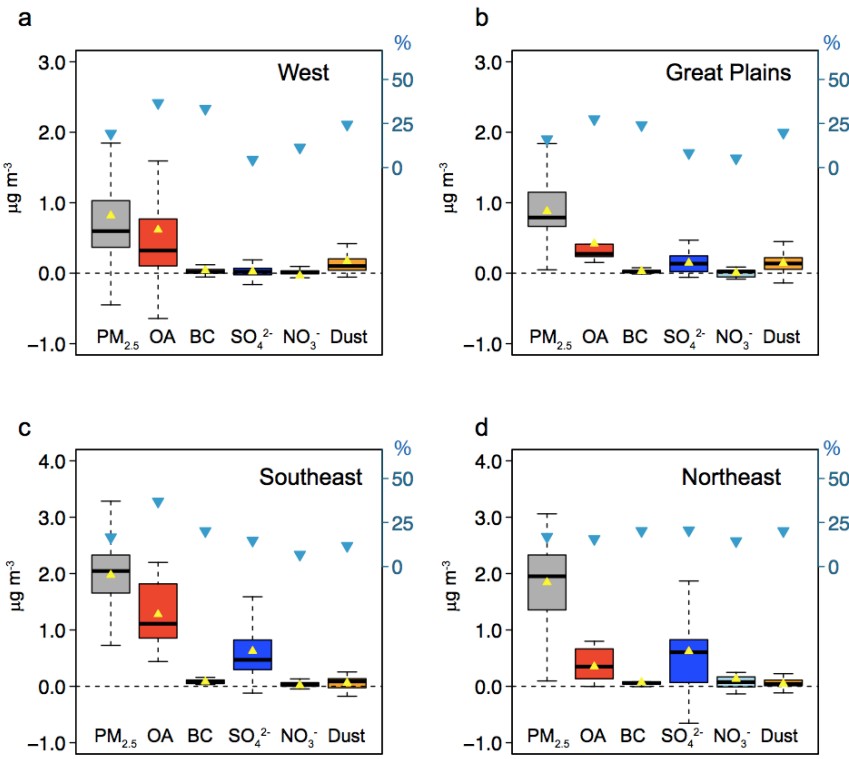

**Figure 5:** Boxplot of anomalies in $PM_{2.5}$ speciation during drought (SPEI < -1.3) compared to normal (-0.5 < SPEI< 0.5) conditions for the Western (a), Great Plains (b), Southeastern (c) and Northeastern (d) US. The yellow triangle indicates mean values and blue triangles indicate relative changes (right y-axis).

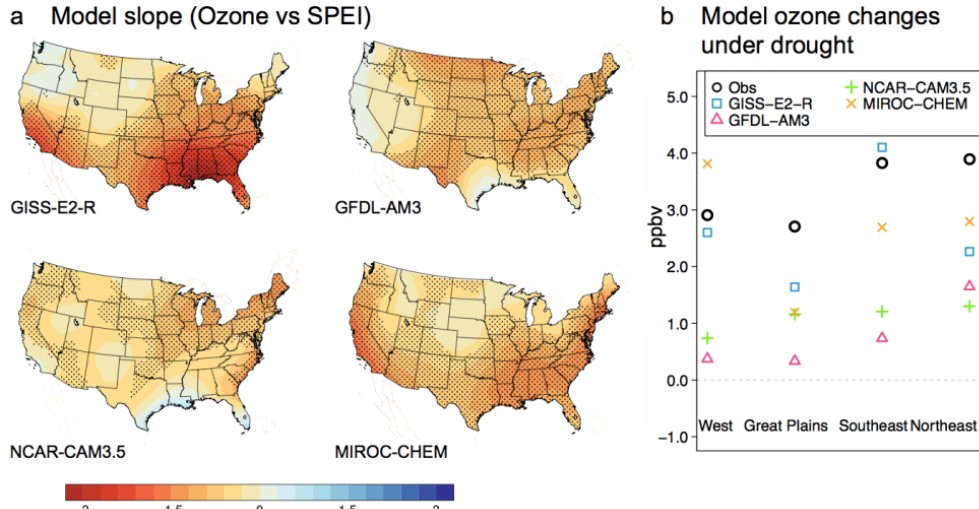

**Figure 6:** Linear regression slope between model derived SPEI and simulated ozone from GISS-E2-R, GFDL-AM3, NCAR-CAM3.5 and MIROC-CHEM model (a). Black dots represent regression significance at 95% confidence level. Note the color bar of (a) is the same as in Figure 1c. Comparison for the observed (black circle) and simulated changes (colored symbols) in ozone (b) under drought (SPEI < -1.3) compared to normal (-0.5 < SPEI < 0.5) condition by region.

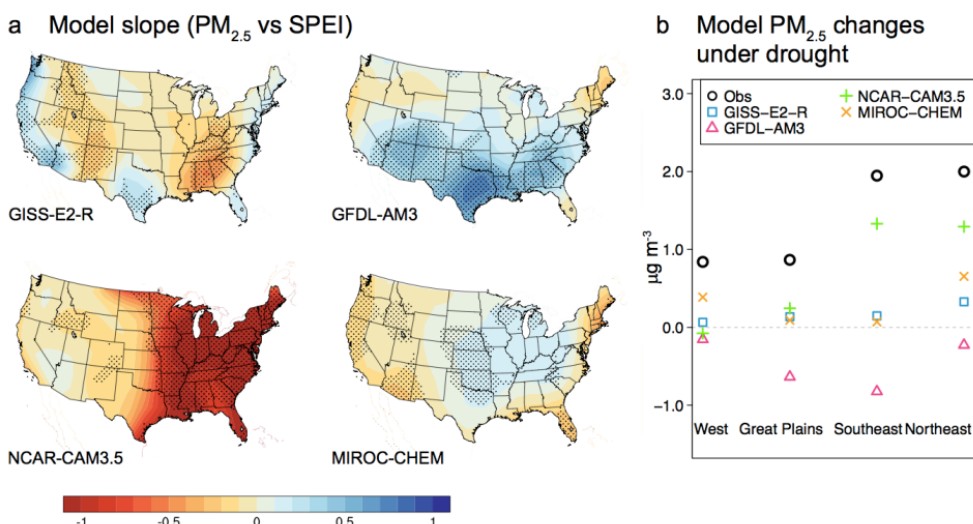

**Figure 7:** Linear regression slope between model derived SPEI and simulated $PM_{2.5}$ from GISS-E2-R, GFDL-AM3, NCAR-CAM3.5 and MIROC-CHEM model (a). Black dots represent regression significance at 95% confidence level. Note the color bar of (a) is the same as in Figure 1e. Comparison for the observed (black circle) and simulated changes (colored symbols) in $PM_{2.5}$ (b) under drought (SPEI < -1.3) compared to normal (-0.5 < SPEI < 0.5) condition by region.

a) Slope between SPEI and total cloud fraction

Observation (ISCCP-D1like)      Model (GISS-E2-R)      Model (GFDL-AM3)

b) Slope between SPEI and boundary layer cloud fraction

Observation (ISCCP-D1like)      Model (GISS-E2-R)      Model (GFDL-AM3)

**Figure 8.** Slopes from linear regression between SPEI and (a) total and (b) boundary layer cloud fractions from the ISCCP satellite observations (left), GISS-E2-R (middle) and GFDL-AM3 model (right)

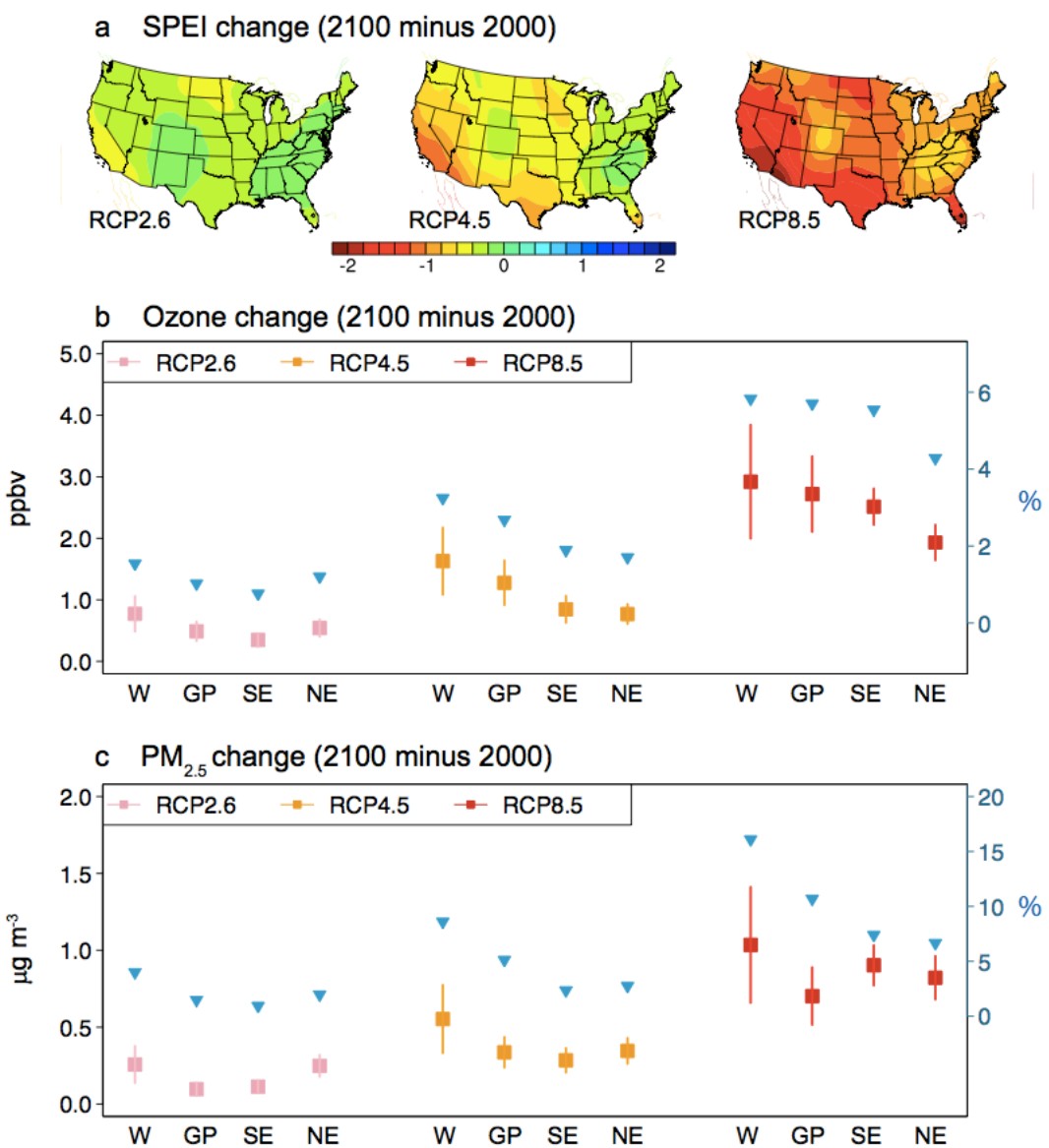

**Figure 9:** Predicted changes in SPEI between 2100 (2080-2099 average) and 2000 (1990-2005 average) by region under three RCP scenarios (a) from mean of four models (GISS-E2-R, GFDL-CM3, CCSM4 and MIROC-ESM-CHEM) and the estimated changes in surface ozone (b) and $PM_{2.5}$ (c) resulting from the SPEI changes alone. The four points in each RCP scenario represent the Western, Great Plains, Southeastern and Northeastern US. Error bar represents 1/2 standard deviation. Blue triangles indicate the mean percentage changes relative to the 2000 conditions (right y-axis).