# Peer review of "Adverse Effects of Increasing Drought on Air Quality via Natural Processes"

_Atmospheric Chemistry and Physics, 2017_

## Referee Comment (RC1) · Anonymous Referee #1 · 1 May 2017

This study addresses the effects of drought on air quality in the United States through statistical analysis of historical observations at surface monitoring sites and two drought indices, the Standardized Precipitation Evaporation Index (SPEI) and the Palmer Drought Severity Index (PDSI). it also examines the ability of several current climate-chemistry models to simulate observed responses of ozone and fine particulate matter under drought conditions as identified by model-derived SPEIs. Future model projections of SPEI and air quality are examined as well. The relationship of drought and air quality is a timely, highly relevant topic, appropriate for the readership of Atmospheric Chemistry and Physics.

Overall, the manuscript is generally well-written with only a few minor typographical or grammatical areas. There are several technical questions/comments that should be addressed prior to reconsideration for publication:

[Figure]

(1) Have many previous studies examined relationships between drought indices and observed air quality? Previous studies should be identified and to the extent possible discussed in the context of this work. See for example, Tian et al. doi: 10.1002/ehs2.1203.

(2) Many drought indices exist now and the number will likely further evolve in the future. Are there indices that are particularly relevant for examining the relationship between drought and air quality, and if so why?

(3) Is there evidence in the historical data that the timing of the onset of drought influences air quality (e.g., late spring vs. early summer vs. late summer)? Is there evidence that prolonged drought more strongly influences air quality over time?

(4) More explanation as to how model-derived SPEIs were calculated (e.g. what method in the R package was used to determine PET?) and their performance relative to the global SPEI dataset and to each other would be beneficial. Model-derived SPEIs are important to establishing predicted air quality during drought versus non-drought conditions and evaluating model deficiencies relative to observed responses in this work.

(5) It is acknowledged in the manuscript that the ACCMIP models vary widely in their predicted responses of air quality to drought. More explanation is needed regarding differences in the configuration and input data resources that could contribute to differences in their performance. A key outcome of this study should be to recommend specific paths forward for research that could lead to improvements in chemistry-climate model performance.

(6) Table 1, Fig. 2, Table S2 etc suggest that there are regional differences in contributions to drought effects to air quality, but the discussion is too limited in this regard. Are there opportunities to better understand model performance via examining regional responses?

Minor corrections: First paragraph, introduction: Line 2: "matters" should be matter; Line 4: missing "the" at the of the line; Line 10: missing noun after "recurring", Line 11: missing "the" before "atmosphere". Page 6, Line 2: "primarily resulted from" should be "primarily result from"
* * *

---

## Referee Comment (RC2) · Anonymous Referee #2 · 23 May 2017

While I like the research topic and design of this paper very much and think that it's long overdue for a comprehensive paper on understanding the effects of drought on air quality, some of the main statements are too broad. Several related issues were mentioned in the initial review. If changes were made, it would have made this review simpler.

I will use the abstract for illustrations. A statement such as "These enhancements show little sensitivity to the decreasing trend of US anthropogenic emissions. . ." is too strong. The decrease of sulfate has been very significant in the US. It is hard to believe that this decrease has no effect on an analysis of the effects of drought on sulfate (or PM2.5 in general). In this paper, there is no comparison of the general decreasing trend of sulfate due to emission reduction with the effects of drought to support the statement.

[Figure]

Another statement in the abstract "Most climate-chemistry models are not able to reproduce the observed responses of ozone and PM2.5 to drought severity, suggesting a lack of mechanistic understanding of drought effects on atmospheric composition." The results from this paper show that there are deficiencies in climate-chemistry models. These deficiencies do not necessary imply that there are missing mechanisms in the models. In the discussion section of the paper (which I like better than the abstract), uncertainties in the models were described. It seems to me that the model deficiencies are more a problem of model representation of drought events not that there is clear evidence for missing climate-chemistry mechanisms.

The statement "Drought thus poses another aspect of climate change penalty on air quality not recognized before" would imply that there were no studies of the kind before. Wang and Xie et al. (2015), for example, obviously discussed some of the issues.

My understanding of the "lack of mechanistic understanding" that the authors referred to is that it is more an issue of how to diagnose the reasons that the model cannot reproduce the observed effects of drought on ozone and aerosols. Drought affects pollutant concentrations through meteorological processes represented by variables such T, RH, and wind speed. there are many papers discussing these "mechanisms". Some are already referenced. But as I wrote in the initial review, the more relevant recent papers were not referenced. Zhang and Wang (2016) discussed the collinear problem in the correlations of ozone with T and RH (as was also seen in this paper) and ozone sensitivities to isoprene emissions. Ozone is much more sensitive to isoprene emissions in the fall than the summer. It is obviously relevant to the discussion of isoprene emissions in this paper.

Zhang et al. (2017) showed that ozone high extremes are more likely to co-occur with high T and low RH. But they also showed that PM2.5 high extremes co-occur with high T and low wind speed but do not depend as much on RH in spring and fall. Drought events have high T, low RH, and low wind speed. Therefore, a drought index, which is more related to RH and T than wind speed, is not the most optimal variable to define

the effects of meteorology on PM2.5 in seasons other than summer. The model errors that the authors referred to may be related to model biases in wind speed simulations under high T and low RH conditions (i.e. simulated drought).

Based on Zhang et al. (2017), I suspect that ozone concentrations have a better correlation with a drought index than PM2.5 in spring and fall, even though the slope seems steeper for PM2.5 than ozone as a function of a drought index. This is obviously important in how the regression slopes can be used to infer the effects of drought. I already suggested in the initial review that the authors include a figure for the distributions of correlation coefficients of ozone and PM2.5 with a drought index (akin to Fig. 1). I make the same recommendation here.

It would take another paper to sort out all the details of why drought conditions lead to higher ozone and PM2.5 concentrations. That is not what I suggest that the authors do in this paper. But the relevant discussion suggested above should be included in the paper. Grouping data in summer with those in spring and fall is not a good choice (Zhang et al., 2017). Analyzing the data in summer and spring+fall separately will be much better. It may be a large amount of work, so I leave the choice to the authors. The authors may choose not to redo the analyses based on season. It is fine with this reviewer as long as the discussion of this seasonal issue is added.

References

Zhang, H., Y. Wang, T.-W. Park, and Y. Deng, Quantifying the relationship between extreme air pollution events and extreme weather events, Atmos. Res., 188, 64-79, doi:10.1016/j.atmosres.2016.11.010, 2017.

Zhang, Y., and Y. Wang, Climate driven ground-level ozone extreme in the fall over the Southeast United States, Proc. Natl. Acad. Sci., 113, 10025–10030, doi: 10.1073/pnas.1602563113, 2016.

---

## Referee Comment (RC3) · Anonymous Referee #3 · 1 Jun 2017

This study examines the correlations between drought and air pollutants (ozone and PM2.5) in the US. The authors use the linear regression slope derived from drought indices and ozone/PM observations to infer the effects of drought and argue that most chemistry-climate models are not able to reproduce the observed relationships. The authors further apply the observed relationships to climate model projected drought occurrences and attempt to estimate the effects of increasing drought on ozone and PM by 2100 compared to the 2000s.

The manuscript is well structured and readable. However, there is a major flaw in the applied method to quantify the impact of drought. The correlations between drought and ozone reported in this study may reflect the common underlying correlation with air stagnation and temperature rather than a causal relationship of drought on ozone. An inspection of the model differences in their Figure 5 supports this statement. None of

these models include the effects of soil moisture deficits on BVOC emissions and the reduced efficiency of ozone dry deposition sink to vegetation. Nevertheless, the GISS model with interactive isoprene emissions simulates the SPEI/ozone slope comparable to the observed values over the Southeast. The greater slope simulated in GISS as compared to other models reflects the inclusion of interactive isoprene emissions, which allows the model to simulate ozone enhancements resulting from stronger isoprene emissions during heat waves (see Schnell et al., 2016). Reduced BVOC emissions under drought stress will actually lead to less ozone.

While severe drought can potentially lead to elevated surface ozone by reducing the ozone dry deposition sink to vegetation (see a review by Fowler et al., 2009), this impact has to be demonstrated using a more sophisticated statistical approach (e.g., multi-variate regression) or chemistry-climate model sensitivity experiments to isolate the role of air stagnation and temperature. For example, Lin et al. (2017) showed that reducing ozone Vd by 35% in GFDL-AM3 during the severe North American drought of 1988 simulates 10 ppbv greater ozone enhancements than a BASE simulation with constant Vd, although the BASE simulation still captures observed ozone enhancements during the other warm summers driven by processes other than drought (see their Section 6 and Figs.18 and 19).

Without a more careful attribution analysis to separate the influence of stagnation and temperature, you cannot use the terms like "drought-induced", "causes of ozone and PM enhancements by drought" or "effects of droughts". All such terms in the present manuscript will need to be removed or rephrased.

In summary, the analysis presented in the current manuscript shows a correlation between drought indices and air pollutants but not the causal effects of drought on air quality. The derived slope may serve as a useful diagnostic to evaluate the models, as the authors show, but it cannot be used to quantify the impact of drought on air quality.

Relevant references:

[Figure]

Fowler, D., et al. (2009),ÂăAtmospheric composition change: Ecosystems–atmosphere interactions,ÂăAtmos. Environ., doi:10.1016/j.atmosenv.2009.07.068.

Lin, M.Y., W. Horowitz, R. Payton, A.M. Fiore, G. Tonnesen (2017).ÂăUS surface ozone trends and extremes from 1980 to 2014: Quantifying the roles of rising Asian emissions, domestic controls, wildfires, and climate.ÂăAtmos. Chem. Phys.,Âădoi:10.5194/acp-17-2943-2017

Schnell, J. L., Prather, M. J., Josse, B., Naik, V., Horowitz, L. W., Zeng, G., Shindell, D. T., and Faluvegi, G.: Effect of climate change on surface ozone over North America, Eu- rope, and East Asia, Geophys. Res. Lett., 43, 3509–3518, doi:10.1002/2016GL068060, 2016.

Sarah C. Kavassalis, Jennifer G. Murphy: Understanding ozone-meteorology correlations: A role for dry deposition, Geophysical Research Letter, 2017

---

## Author Comment (AC1) · 5 Aug 2017

**Response to Reviews**

We thank the reviewer for constructive comments to improve the manuscript. The comments are reproduced below with our responses in blue. The corresponding changes in the manuscript are highlighted in blue.

**Reviewer #1**

This study addresses the effects of drought on air quality in the United States through statistical analysis of historical observations at surface monitoring sites and two drought indices, the Standardized Precipitation Evaporation Index (SPEI) and the Palmer Drought Severity Index (PDSI). It also examines the ability of several current climate-chemistry models to simulate observed responses of ozone and fine particulate matter under drought conditions as identified by model-derived SPEIs. Future model projections of SPEI and air quality are examined as well. The relationship of drought and air quality is a timely, highly relevant topic, appropriate for the readership of Atmospheric Chemistry and Physics.

Overall, the manuscript is generally well-written with only a few minor typographical or grammatical areas. There are several technical questions/comments that should be addressed prior to reconsideration for publication:

(1) Have many previous studies examined relationships between drought indices and observed air quality? Previous studies should be identified and to the extent possible discussed in the context of this work. See for example, Tian et al. doi:10.1002/ehs2.1203.

To our knowledge, few studies have examined the relationship between drought indices and observed air quality at a temporal and spatial scale similar to our study (i.e. 25 years, continental US). There are a few papers analyzing on one or two aspects of the drought impact on atmospheric compositions associated with dust and fire emissions (Prospero and Lamb, 2003;Westerling and Swetnam, 2003). Tian et al (2016) analyzed the combined effects of drought and ozone on crop productions in China, but they did not explicitly consider the drought effects on ozone. Our previous work (Wang et al., 2015) conducted a case study of surface $PM_{2.5}$ enhancements associated with the 2011 southern US drought. We have added discussions of all these previous studies in the introduction of the revised manuscript.

(2) Many drought indices exist now and the number will likely further evolve in the future. Are there indices that are particularly relevant for examining the relationship between drought and air quality, and if so why?

Air quality responds to changes not only of the atmosphere but also the land biosphere, thus the drought indices that are most relevant for air quality would be those that measure both meteorological (e.g., temperature and precipitation) and land biosphere conditions (e.g., soil moisture, evapotranspiration, vegetation, etc.)

associated with drought. In addition, the temporal duration of drought is a matter of concern for air quality because air pollutants have different characteristic time scales with respect to transport and chemistry. This requires the relevant drought indices to be explicit of drought duration (e.g., month, year) in their calculation.

Take the Standardized Precipitation Evapotranspiration Index (SPEI) as an example, which is the primary drought index used in our study. The SPEI is based on water balance between precipitation and reference evapotranspiration, the latter dependent on atmospheric water demand related to temperature. Therefore it represents both meteorological conditions and water stress on land biosphere conditions during drought. In addition, the SPEI is multi-temporal and can specify drought duration of monthly, and multi-months. Our study used the 1-month SPEI and the correspondent monthly-mean air pollutant data (ozone and PM2.5) to derive the relationship between drought and air quality. By comparison, the Standard Precipitation Index (SPI) or the PDSI would not be a good drought index for air quality purpose because the SPI considers only meteorology (i.e. precipitation) while the PDSI does not specify drought duration.

The above points were implicit in the original manuscript where the SPEI is introduced (Section 2.1). We've now explicitly expressed them in the revised Section 2.1.

(3) Is there evidence in the historical data that the timing of the onset of drought influences air quality (e.g., late spring vs. early summer vs. late summer)? Is there evidence that prolonged drought more strongly influences air quality over time?

We added a new Figure 2 comparing the different effects of drought onset and prolonged drought on ozone and $PM_{2.5}$ enhancements. Both pollutants show larger enhancements during prolonged drought compared to drought onset across the four regions, except for $PM_{2.5}$ over the northeast. See the detailed discussion about Figure 2 added at the end of Section 3.1.

(4) More explanation as to how model-derived SPEIs were calculated (e.g. what method in the R package was used to determine PET?) and their performance relative to the global SPEI dataset and to each other would be beneficial. Model-derived SPEIs are important to establishing predicted air quality during drought versus nondrought conditions and evaluating model deficiencies relative to observed responses in this work.

The model-derived SPEI were calculated with R package provided by the SPEI developer using model precipitation and temperature as inputs. The SPEI is derived as logistic-normalized distribution of water deficit, estimated as the difference between precipitation and reference evapotranspiration. Both Thornthwaite (Th) and Penman-Monteith (PM) method can be applied for estimation of the reference evapotranspiration. The Thornthwaite (Th) method only requires temperature data while the Penman-Monteith (PM) method requires additional inputs including RH,

wind speed and radiation. Since ACCMIP model archives do not have all the variables required for the PM method, we used the Th method to calculate model SPEI. The global SPEI dataset use the PM method to estimate reference evapotranspiration. The correlation between SPEIs derived with PM and Th method is high (correlation r >0.9) (Beguera et al., 2014), thus the use of Th method may not have large impact on model SPEI calculation. We've clarified this point in the manuscript (Section 2.3).

With regard to the model ability of simulating drought, Figure S7 in the supplementary material presents the model-simulated drought frequencies during the historical period (1990-2014). The models can capture well the observed spatial patterns of drought occurrence frequency. Severe drought (model SPEI < -1.3) occurs ~20% of the time over the west and southern US, consistent with observed SPEI. However, the temporal correspondence (i.e. month-to-month) between model SPEI and global SPEI dataset is weak, largely due to the models deficiency in simulating temporal variability of precipitation. This weak correlation however is not expected to affect the model evaluation because we used the model SPEI to derive the simulated SPEI-pollutants relationships from each model. We've added discussion of the model SPEI in the manuscript (pg 9, line 33-39).

(5) It is acknowledged in the manuscript that the ACCMIP models vary widely in their predicted responses of air quality to drought. More explanation is needed regarding differences in the configuration and input data resources that could contribute to differences in their performance. A key outcome of this study should be to recommend specific paths forward for research that could lead to improvements in chemistry-climate model performance.

Agreed. Emissions, deposition, and chemistry are the most important aspects of model configurations affecting the drought-pollutants relationship. Anthropogenic emissions and biomass burning emissions were specified, but natural emissions were not, so the models treated natural emissions differently, which is a key factor in the different performance between models. For example, only the GISS-E2-R model simulates isoprene emissions as coupled with its meteorology (mostly temperature), thus allowing for isoprene emissions to increase with increasing temperatures. The other three models used prescribed BVOC emissions, thus representing different responses of those emissions to meteorology and climate change. All the ACCMIP models include dry and wet deposition of pollutants. While they all show large reductions of wet deposition during drought, the dry deposition is not sensitive to drought. With regard to aerosol chemistry, all the models overestimate the sulfate reduction, but at the same time underestimate the OA increase during drought. Both problems might be caused by the model misrepresentation of cloud sensitivity to changing drought severity, although the OA bias could also be caused by uncertainties of fire and BVOC emissions in the models. We've expanded the modeling discussion in Section 3.4 (last paragraph).

(6) Table 1, Fig. 2, Table S2 etc suggest that there are regional differences in

contributions to drought effects to air quality, but the discussion is too limited in this regard. Are there opportunities to better understand model performance via examining regional responses?

This is a good point. We've expanded the discussion of regional differences in the revised Section 3.1 when presenting the regional-mean pollutants enhancements associated with drought (e.g. new Figure 2), as well as in the newly added Section 3.2 when presenting regional differences in meteorology during drought (e.g. new Figure 3 and Figure S4). A detailed region-to-region comparison is however outside the scope of the current manuscript and will be a future endeavor, as our main goal here is to provide observational evidence of the robustness and spatial prevalence of pollution enhancements during drought across the US.

Minor corrections:
First paragraph, introduction: Line 2: "matters" should be matter;
Line 4: missing "the" at the of the line; Line 10: missing noun after "recurring", Line 11: missing "the" before "atmosphere". Page 6, Line 2: "primarily resulted from" should be "primarily result from"

All are corrected.

References:

Beguera, S., Vicente‐Serrano, S. M., Reig, F., and Latorre, B.: Standardized precipitation evapotranspiration index (SPEI) revisited: parameter fitting, evapotranspiration models, tools, datasets and drought monitoring, *International Journal of Climatology*, 34, 3001-3023, 2014.

Prospero, J. M., and Lamb, P. J.: African droughts and dust transport to the Caribbean: Climate change implications, *Science*, 302, 1024-1027, 2003.

Tian, H., Ren, W., Tao, B., Sun, G., Chappelka, A., Wang, X., Pan, S., Yang, J., Liu, J., and S Felzer, B.: Climate extremes and ozone pollution: a growing threat to China's food security, *Ecosystem Health and Sustainability,* 2, 2016.

Wang, Y., Xie, Y., Cai, L., Dong, W., Zhang, Q., and Zhang, L.: Impact of the 2011 southern US drought on ground-level fine aerosol concentration in summertime, *Journal of the Atmospheric Sciences*, 72, 1075-1093, 2015.

Westerling, A. L., and Swetnam, T. W.: Interannual to decadal drought and wildfire in the western United States, *EOS, Transactions American Geophysical Union*, 84, 545-555, 2003.

---

## Author Comment (AC2) · 5 Aug 2017

**Response to Reviews**

We thank the reviewer for constructive comments to improve the manuscript. The comments are reproduced below with our responses in blue. The corresponding changes in the manuscript are highlighted in blue.

**Reviewer #2**

While I like the research topic and design of this paper very much and think that it's long overdue for a comprehensive paper on understanding the effects of drought on air quality, some of the main statements are too broad. Several related issues were mentioned in the initial review. If changes were made, it would have made this review simpler.

The reviewer made a comment in his/her initial review about using different cloud fractions (CF) for ozone and aerosols, which we chose to address here in the final revision stage. We agree with the reviewer that total cloud fraction should be used for the analysis of the radiation effects on ozone chemistry, while boundary layer cloud fraction should be used for the analysis of in-cloud oxidation of $SO_2$. In the revised Figure 8 and Table S2, we have added both total cloud fraction (integrated between 1000 and 10 hPa) and boundary layer cloud fraction (integrated between 1000 and 800 hPa) from the ISCCP observation and the two models (GDFL and GISS) that archived layer specific cloud fractions. The ISCCP data show a 9-24% decrease in total CF and 6-13% decrease in boundary layer CF during drought periods (Table S2). Both the GFDL and GISS model show much larger (30-47%) decreases of total and boundary layer CF. The correlation slope between SPEI and total/boundary layer CF is about 10 times higher in the two models than that from observations (Figure 8). This confirms our original finding that the models tend to underestimate cloud fractions (both total CF and boundary CF) during drought, leading to excessive reductions of in-cloud formation of sulfate aerosols.

I will use the abstract for illustrations. A statement such as "These enhancements show little sensitivity to the decreasing trend of US anthropogenic emissions: : :" is too strong. The decrease of sulfate has been very significant in the US. It is hard to believe that this decrease has no effect on an analysis of the effects of drought on sulfate (or PM2.5 in general). In this paper, there is no comparison of the general decreasing trend of sulfate due to emission reduction with the effects of drought to support the statement.

We agree with the reviewer that a few statements need to be revised to improve clarity and specify scope. The statement in question here does not refer to the actual concentrations of ozone or $PM_{2.5}$, which indeed show a large decrease over 1990-2014 with decreasing US anthropogenic emissions (see Table 2, last column). We meant to say that the pollutant enhancements associated with droughts do not change at the same rate or even the same direction of decreasing anthropogenic emissions in the US. We've revised the statement as: "The pollutant enhancements

associated with droughts do not appear to be affected by the decreasing trend of US anthropogenic emissions, indicating natural processes as the primary cause".

Another statement in the abstract "Most climate-chemistry models are not able to reproduce the observed responses of ozone and PM2.5 to drought severity, suggesting a lack of mechanistic understanding of drought effects on atmospheric composition." The results from this paper show that there are deficiencies in climate-chemistry models. These deficiencies do not necessary imply that there are missing mechanisms in the models. In the discussion section of the paper (which I like better than the abstract), uncertainties in the models were described. It seems to me that the model deficiencies are more a problem of model representation of drought events not that there is clear evidence for missing climate-chemistry mechanisms.

Agreed. We've removed the second part of the sentence (i.e. removed "suggesting a lack of ..."). Regarding the models' ability to simulate drought, we showed in Figure S7 (Supplementary Material) that the four ACCMIP models were able to capture the observed spatial patterns of drought occurrence frequency. Severe drought (model SPEI < -1.3) occurs ~20% of the time over the west and southern US, consistent with observed SPEI. However, the temporal correspondence (i.e. month-to-month) between model SPEI and global SPEI dataset is weak, largely due to the models deficiency in simulating temporal variability of precipitation. This weak correlation however is not expected to affect the model evaluation, because we used the model SPEI to derive the simulated SPEI-pollutants relationships from each model and compared those relationships between model and observations, rather than ozone or PM2.5 concentrations per se. We've added discussion of the model ability to simulate drought in the manuscript (pg 9, line 33-39).

The statement "Drought thus poses another aspect of climate change penalty on air quality not recognized before" would imply that there were no studies of the kind before. Wang and Xie et al. (2015), for example, obviously discussed some of the issues.

Agreed. In fact, the work of Wang and Xie et al. (2015) referred by the reviewer is our own. The part "not recognized before" is removed from that sentence.

My understanding of the "lack of mechanistic understanding" that the authors referred to is that it is more an issue of how to diagnose the reasons that the model cannot reproduce the observed effects of drought on ozone and aerosols. Drought affects pollutant concentrations through meteorological processes represented by variables such T, RH, and wind speed. There are many papers discussing these "mechanisms". Some are already referenced. But as I wrote in the initial review, the more relevant recent papers were not referenced. Zhang and Wang (2016) discussed the collinear problem in the correlations of ozone with T and RH (as was also seen in this paper) and ozone sensitivities to isoprene emissions. Ozone is much more sensitive to isoprene emissions in the fall than the summer. It is obviously relevant to the discussion of isoprene emissions in this paper.

The reviewer's point is well taken. We've added a new section (Section 3.2) to discuss extensively the meteorological factors responsible for the drought-pollutant relationship, such as temperature, RH, and wind speed. More relevant recent papers have been added as references, including Zhang and Wang (2016) and Zhang et al. (2017) mentioned by the reviewer. We acknowledge in this new section that there are well-established linkages between air quality and some meteorological parameters (e.g. temperature), thus the drought-pollution association may be partly explained by the effects of drought on these meteorological variables.

Zhang et al. (2017) showed that ozone high extremes are more likely to co-occur with high T and low RH. But they also showed that PM2.5 high extremes co-occur with high T and low wind speed but do not depend as much on RH in spring and fall. Drought events have high T, low RH, and low wind speed. Therefore, a drought index, which is more related to RH and T than wind speed, is not the most optimal variable to define the effects of meteorology on PM2.5 in seasons other than summer. The model errors that the authors referred to may be related to model biases in wind speed simulations under high T and low RH conditions (i.e. simulated drought).

In the new section 3.2, we added discussions of the differences between drought and meteorological factors (temperature, RH, and winds) and other meteorological events, including heat wave and stagnation, which are associated with high pollution levels and likely co-occur with drought. The first difference is that drought is not a daily-scale extreme or variable, such as temperature or RH. Drought arises only after a prolonged (> week) period of precipitation shortage that causes soil to dry up. Therefore, we chose the monthly scale to identify the drought-pollution association, differentiating it from day-to-day variability of meteorology. Second, drought is a complex extreme not based on individual meteorological parameters (e.g. temperature, humidity) or a simple combination of them. The prominent feature of drought is water deficit in both the atmosphere and the land component (e.g. soil and vegetation), resulting from the combination of precipitation shortage and increasing evapotranspirative water loss driven in part by high temperatures. As a result, the associated vegetation responses are likely to be more pronounced during drought than those associated with short-term meteorological extremes/events, which are relevant to our later discussion of isoprene changes.

We have added a clear statement to acknowledge that the co-occurrence of high temperature and low RH with drought is an important reason to explain the pollutant enhancements during drought, especially for surface ozone. However, it would not be feasible to separately quantify the effects of certain meteorological variables on the drought-pollution association, such as temperature, precipitation, and RH, because these variables are all factored in when defining drought. But wind speed is not an explicit factor in drought indices, thus we can evaluate if wind is a compounding meteorological factor for the drought-pollution association. The correlations (r) of monthly mean wind speeds with the SPEI (see new Figure S4 in supplementary material) are positive but small for the most part of the US ($r^2 < 0.2$). This suggests that wind speeds might not be an important meteorological factor responsible for the pollution enhancements during drought, except for localized

areas where wind-blowing dust would be substantially higher during drought.

Based on Zhang et al. (2017), I suspect that ozone concentrations have a better correlation with a drought index than PM2.5 in spring and fall, even though the slope seems steeper for PM2.5 than ozone as a function of a drought index. This is obviously important in how the regression slopes can be used to infer the effects of drought. I already suggested in the initial review that the authors include a figure for the distributions of correlation coefficients of ozone and PM2.5 with a drought index (akin to Fig. 1). I make the same recommendation here.

The original manuscript showed the distributions of correlation coefficients in the supplementary material. We've now moved those figures to the main manuscript (new Figure 1). The correlation coefficients have similar spatial distributions as the correlation slopes for both ozone and $PM_{2.5}$.

It would take another paper to sort out all the details of why drought conditions lead to higher ozone and PM2.5 concentrations. That is not what I suggest that the authors do in this paper. But the relevant discussion suggested above should be included in the paper. Grouping data in summer with those in spring and fall is not a good choice (Zhang et al., 2017). Analyzing the data in summer and spring+fall separately will be much better. It may be a large amount of work, so I leave the choice to the authors. The authors may choose not to redo the analyses based on season. It is fine with this reviewer as long as the discussion of this seasonal issue is added.

The reviewer's point is well taken. We've added separate analysis of the ozone and $PM_{2.5}$ enhancement by season (spring, summer, and fall). See the new Figure 2 and related discussion added at the end of the revised Section 3.1. For ozone, all the regions see larger ozone enhancements in summer (Jun-Aug) and fall (Sep-Oct), while the spring (Mar-May) enhancement is the smallest. The seasonal differences of $PM_{2.5}$ enhancements are not statistically significant for most regions, nor are they coherent between regions, probably due to the complexity in $PM_{2.5}$ chemical constituents and sources. The seasonal comparison for a given region is based on the same sets of surface sites that experience droughts in all the seasons, thus the differences presented in the revised manuscript are not caused by sampling differences. The seasonal analysis supports the robustness of the drought-pollution association derived over the growing season as a whole.

References

Zhang, H., Y. Wang, T.-W. Park, and Y. Deng, Quantifying the relationship between extreme air pollution events and extreme weather events, *Atmos. Res.*, 188, 64-79, doi:10.1016/j.atmosres.2016.11.010, 2017.

Zhang, Y., and Y. Wang, Climate driven ground-level ozone extreme in the fall over the Southeast United States, *Proc. Natl. Acad. Sci.*, 113, 10025–10030, doi: 10.1073/pnas.1602563113, 2011

---

## Author Comment (AC3) · 5 Aug 2017

**Response to Reviews**

We thank the reviewer for constructive comments to improve the manuscript. The comments are reproduced below with our responses in blue. The corresponding changes in the manuscript are highlighted in blue.

**Reviewer #3**

This study examines the correlations between drought and air pollutants (ozone and PM2.5) in the US. The authors use the linear regression slope derived from drought indices and ozone/PM observations to infer the effects of drought and argue that most chemistry-climate models are not able to reproduce the observed relationships. The authors further apply the observed relationships to climate model projected drought occurrences and attempt to estimate the effects of increasing drought on ozone and PM by 2100 compared to the 2000s.

The manuscript is well structured and readable. However, there is a major flaw in the applied method to quantify the impact of drought. The correlations between drought and ozone reported in this study may reflect the common underlying correlation with air stagnation and temperature rather than a causal relationship of drought on ozone. An inspection of the model differences in their Figure 5 supports this statement. None of these models include the effects of soil moisture deficits on BVOC emissions and the reduced efficiency of ozone dry deposition sink to vegetation. Nevertheless, the GISS model with interactive isoprene emissions simulates the SPEI/ozone slope comparable to the observed values over the Southeast. The greater slope simulated in GISS as compared to other models reflects the inclusion of interactive isoprene emissions, which allows the model to simulate ozone enhancements resulting from stronger isoprene emissions during heat waves (see Schnell et al., 2016). Reduced BVOC emissions under drought stress will actually lead to less ozone.

We thank the reviewer for making this important point, which we have careful analyzed and extensively discussed in the revised manuscript. We've added a new session (Section 3.2) and several new figures (Figure 3, Figure S4-S6) to discuss the meteorological factors behind the SPEI-pollutant relationship and the possible compounding effects of stagnation and high temperatures.

In the new Section 3.2, we first acknowledge that there are well-established linkages between air quality and some meteorological parameters (e.g. temperature), thus the drought-pollution association may be partly explained by the effects of drought on these meteorological variables. We then discussed the differences between drought and meteorological parameters (temperature, RH, and winds) and meteorological events, including heat wave and stagnation. The first difference is that drought is not a daily-scale extreme or variable, such as temperature or RH. Drought arises only after a prolonged (> week) period of precipitation shortage that causes soil to dry up. Therefore, we chose the monthly scale to identify the drought-pollution association,

differentiating it from day-to-day variability of meteorology. Second, drought is a complex extreme not based on individual meteorological parameters (e.g. temperature, humidity) or a simple combination of them. The prominent feature of drought is water deficit in both the atmosphere and the land component (e.g. soil and vegetation), resulting from the combination of precipitation shortage and increasing evapotranspirative water loss driven in part by high temperatures. As a result, the associated vegetation responses are likely to be more pronounced during drought than those associated with short-term meteorological extremes/events, which are relevant to our later discussion of isoprene changes. The change in isoprene during drought found in our study is consistent with Schnell et al. (2016) in that isoprene tends to increase in most drought conditions (Figure 4); only under extreme drought (SPEI <-2) isoprene is shown to decrease in limited observations.

We then examined the relationships of monthly occurrences of stagnation and heat waves with the SPEI at each $0.5^{o}$x$0.5^{o}$ grid over the study period; the results are shown in the new Figure S4 in the supplementary material. There are positive correlations between drought-stagnation and drought-heatwave across the US, indicating both events do occur more frequently during drought months. But the squares of these correlations are all below 0.4, with a typical value of 0.1-0.2 for the most parts of the US. This suggests that on the monthly scale stagnation and heat waves would typically be able to explain 10%~20% variability in the SPEI, a non-trivial but small fraction. As shown in the lower panel of Figure S4, stagnation and heat waves have an average 7% and 5% increase in their frequencies during drought months compared to normal months, although the extent of such increases varies greatly by region. The maximal increase of stagnation frequency during drought is about 15% in the west, southern Great Plains and southwest, where stagnation tends to occur frequently even during normal conditions. The largest increase of heat waves during drought is about 20% in the southern Great Plains.

Finally, to quantify the compounding effects of stagnation and heat waves on the drought-pollution association, we re-evaluated the SPEI-pollutant relationships by applying weights to each pair of SPEI and pollutant anomalies (ozone and $PM_{2.5}$). The weights are given as the percentages of days in each month (regardless of drought or non-drought) that are neither stagnation nor heat wave, assuming the two events are mutually exclusive which would give an upper bound for the weights. Since the weights are between 0 and 1, the weighting process effectively scales down the magnitude of pollution anomalies in each month. The weighted correlation and enhancements are shown in the new Figure 3. The weighting does not change the sign or statistical significance of the SPEI-pollutant correlations at all the sites, indicating the covariance of drought with stagnation and heat waves might not be the dominant factor causing the drought-pollutant correlations. The weighting however reduces the magnitude of ozone and $PM_{2.5}$ enhancements associated with drought by an average of 40% when both events are combined. This indicates that more frequent stagnation and heat waves could explain up to 40% of the ozone and $PM_{2.5}$

enhancements during drought, a significant but not majority fraction.

While severe drought can potentially lead to elevated surface ozone by reducing the ozone dry deposition sink to vegetation (see a review by Fowler et al., 2009), this impact has to be demonstrated using a more sophisticated statistical approach (e.g., multi-variate regression) or chemistry-climate model sensitivity experiments to isolate the role of air stagnation and temperature. For example, Lin et al. (2017) showed that reducing ozone Vd by 35% in GFDL-AM3 during the severe North American drought of 1988 simulates 10 ppbv greater ozone enhancements than a BASE simulation with constant Vd, although the BASE simulation still captures observed ozone enhancements during the other warm summers driven by processes other than drought (see their Section 6 and Figs.18 and 19).

We agree with the reviewer that this manuscript is not an attribution analysis (e.g. using chemistry-climate model sensitivity experiments), and we have explicitly stated so in the manuscript: "… attribution of the underlying causes would require chemistry-climate model sensitivity experiments, which is outside the scope of the present study" (pg 10, line 33-36). We've added Lin et al. (2017) as reference when discussing the dry deposition effect.

Without a more careful attribution analysis to separate the influence of stagnation and temperature, you cannot use the terms like "drought-induced", "causes of ozone and PM enhancements by drought" or "effects of droughts". All such terms in the present manuscript will need to be removed or rephrased.

Our new analysis (Section 3.2) showed that the drought-pollution relationship is still significant after discounting the influence of stagnation and temperature on air pollution (see the response to the first comment), which supports our argument that droughts affect air quality. However, we agree with the reviewer that our manuscript is not an attribution analysis by design and thus some of the terms need to be modified. We've modified all the attribution terms throughout the manuscript to non-attribution ones such as "pollutant enhancement associated with droughts" or "drought-pollutant relationship" or "drought-pollutant association".

In summary, the analysis presented in the current manuscript shows a correlation between drought indices and air pollutants but not the causal effects of drought on air quality. The derived slope may serve as a useful diagnostic to evaluate the models, as the authors show, but it cannot be used to quantify the impact of drought on air quality.

We agree. As stated above, we've changed attribution statements to non-attribution ones throughout the manuscript.

Relevant references:

Fowler D, Pilegaard K, Sutton M, Ambus P, Raivonen M, Duyzer J, *et al.* Atmospheric composition change: ecosystems–atmosphere interactions. *Atmospheric Environment* 2009, **43**(33): 5193-5267.

Lin, M., Horowitz, L. W., Payton, R., Fiore, A. M., and Tonnesen, G.: US surface ozone trends and extremes from 1980 to 2014: quantifying the roles of rising Asian emissions, domestic controls, wildfires, and climate, *Atmos. Chem. Phys*., 17, 2943-2970, https://doi.org/10.5194/acp-17-2943-2017, 2017.

Schnell JL, Prather MJ, Josse B, Naik V, Horowitz LW, Zeng G*, et al.* Effect of climate change on surface ozone over North America, Europe, and East Asia. *Geophysical Research Letters*, 43(7)**:** 3509-3518, 2016

Kavassalis, S.C. and Murphy, J.G.: Understanding ozone-meteorology correlations: A role for dry deposition. *Geophysical Research Letters*, 44(6), pp.2922-2931, 2017